# Ecology of inorganic sulfur auxiliary metabolism in widespread bacteriophages

Kristopher Kieft[1,14], Zhichao Zhou[1,14], Rika E. Anderson [2], Alison Buchan[3], Barbara J. Campbell [4], Steven J. Hallam [5,6,7,8,9], Matthias Hess [10], Matthew B. Sullivan [11], David A. Walsh[12], Simon Roux[13] & Karthik Anantharaman [1✉]

Microbial sulfur metabolism contributes to biogeochemical cycling on global scales. Sulfur metabolizing microbes are infected by phages that can encode auxiliary metabolic genes (AMGs) to alter sulfur metabolism within host cells but remain poorly characterized. Here we identified 191 phages derived from twelve environments that encoded 227 AMGs for oxidation of sulfur and thiosulfate (*dsrA*, *dsrC/tusE*, *soxC*, *soxD* and *soxYZ*). Evidence for retention of AMGs during niche-differentiation of diverse phage populations provided evidence that auxiliary metabolism imparts measurable fitness benefits to phages with ramifications for ecosystem biogeochemistry. Gene abundance and expression profiles of AMGs suggested significant contributions by phages to sulfur and thiosulfate oxidation in freshwater lakes and oceans, and a sensitive response to changing sulfur concentrations in hydrothermal environments. Overall, our study provides fundamental insights on the distribution, diversity, and ecology of phage auxiliary metabolism associated with sulfur and reinforces the necessity of incorporating viral contributions into biogeochemical configurations.

[1] Department of Bacteriology, University of Wisconsin–Madison, Madison, WI, USA. [2] Biology Department, Carleton College, Northfield, MN, USA. [3] Department of Microbiology, University of Tennessee, Knoxville, TN, USA. [4] Department of Biological Sciences, Life Science Facility, Clemson University, Clemson, SC, USA. [5] Department of Microbiology & Immunology, University of British Columbia, Vancouver, BC, Canada. [6] Graduate Program in Bioinformatics, University of British Columbia, Genome Sciences Centre, Vancouver, BC, Canada. [7] Genome Science and Technology Program, University of British Columbia, Vancouver, BC, Canada. [8] Life Sciences Institute, University of British Columbia, Vancouver, BC, Canada. [9] ECOSCOPE Training Program, University of British Columbia, Vancouver, BC, Canada. [10] Department of Animal Science, University of California Davis, Davis, CA, USA. [11] Department of Microbiology, The Ohio State University, Columbus, OH, USA. [12] Groupe de recherche interuniversitaire en limnologie, Department of Biology, Concordia University, Montréal, QC, Canada. [13] DOE Joint Genome Institute, Lawrence Berkeley National Laboratory, Berkeley, CA, USA. [14]These authors contributed equally: Kristopher Kieft, Zhichao Zhou. ✉email: karthik@bact.wisc.edu

Viruses that infect bacteria (bacteriophages, or phages) are estimated to encode a larger repertoire of genetic capabilities than their bacterial hosts and are prolific at transferring genes throughout microbial communities[1–4]. The majority of known phages have evolved compact genomes by minimizing non-coding regions, reducing the average length of encoded proteins, fusing proteins and retaining few non-essential genes[5,6]. Despite their reduced genome size and limited coding capacity, phages are known for their ability to modulate host cells during infection, take over cellular metabolic processes and proliferate through a bacterial population, typically through lysis of host cells[7,8]. Phage-infected hosts, termed virocells, take on a distinct physiology compared to an uninfected state[9]. According to some estimates, as many as 20–40% of all bacteria in aquatic environments are assumed to be in a virocell state, undergoing phage-directed metabolism[10,11]. This has led to substantial interest in understanding the mechanisms that provide phages with the ability to redirect nutrients within a host and ultimately how this manipulation may affect microbiomes and ecosystems.

One such mechanism by which phages can alter the metabolic state of their host is through the activity of phage-encoded auxiliary metabolic genes (AMGs)[12,13]. AMGs are typically acquired from the host cell (i.e., recombined onto the phage genome) and can be utilized during infection to augment or redirect specific metabolic processes within the host cell[14–16]. These augmentations likely function to maintain, drive or short-circuit important steps of a metabolic pathway and can provide the phage with sufficient fitness advantages under specific metabolic or nutrient conditions in order to retain these genes over time[12,17]. Two notable examples of AMGs are core photosystem II proteins psbA and psbD, which are commonly encoded by phages infecting Cyanobacteria in both freshwater and marine environments, and responsible for supplementing photosystem function in virocells during infection[18–21]. PsbA and PsbD play important roles in maintenance of photosynthetic energy production over time within the host; this energy is subsequently utilized for the production of resources (e.g., nucleotides) for phage propagation[12,14]. The Cyanobacteria host does not benefit from the additional gene copy (i.e., phage AMG) since the replication benefits and energy acquisition are in favor of the infecting phage. Other descriptions of AMGs include those for sulfur oxidation in the pelagic oceans[16,22], methane oxidation in freshwater lakes[23], ammonia oxidation in surface oceans[24], carbon utilization (e.g., carbohydrate hydrolysis) in soils[25,26], and marine ammonification[27]. As a further example, it has been hypothesized that some phages encoding carbon utilization AMGs function to redirect carbon from glycolysis to deoxynucleotide triphosphate (dNTP) synthesis for phage genome replication, by inducing a state of host starvation[7]. In this scenario, the phages encode their own AMG for specific manipulation of host processes rather than simply providing an extra gene copy to the host. Beyond these examples, the combined effect of phage auxiliary metabolism on ecosystems scales has yet to be fully explored or implemented into conceptualizations of microbial community functions and interactions.

Dissimilatory sulfur metabolism (DSM) encompasses both reduction (e.g., sulfate to sulfide) and oxidation (e.g., sulfide or thiosulfate to sulfate) and accounts for the majority of sulfur metabolism on Earth[28]. Bacteria capable of DSM (termed as sulfur microbes) are phylogenetically diverse, spanning 13 separate phyla, and can be identified throughout a range of natural and human systems, aquatic and terrestrial biomes, aerobic or anaerobic environments, and in the light or dark[29]. Since DSM is often coupled with primary production and the turnover of buried organic carbon, understanding these processes is essential for interpreting the biogeochemical significance of both microbial- and phage-mediated nutrient and energy transformations[29]. Phages of DSM-mediating microorganisms are not well characterized beyond the descriptions of phages encoding dsrA and dsrC genes infecting known sulfur oxidizers from the SUP05 group of Gammaproteobacteria[16,22], and viruses encoding dsrC and soxYZ genes associated with proteobacterial hosts in the epipelagic ocean[30]. Despite the identification of DSM AMGs across multiple host groups and environments, there remains little context for their global diversity and roles in the biogeochemical cycling of sulfur. Characterizing the ecology, function and roles of phages associated with DSM is crucial to an integral understanding of the mechanisms by which sulfur species are transformed and metabolized.

Here, we leveraged publicly available metagenomic and metatranscriptomic data to identify phages capable of manipulating DSM within host cells. We identified 191 phages encoding AMGs for oxidation and disproportionation of reduced sulfur species, such as elemental sulfur and thiosulfate, in coastal ocean, pelagic ocean, hydrothermal vent, human, and terrestrial environments. We refer to these phages encoding AMGs for DSM as 'sulfur phages'. These sulfur phages represent different taxonomic clades of Caudovirales, namely from the families Siphoviridae, Myoviridae and Podoviridae, with diverse gene contents, and evolutionary history. Using paired viral-host gene coverage measurements from metagenomes recovered from hydrothermal environments, freshwater lakes, and Tara Ocean samples, we provide evidence for the significant contribution of viral AMGs to sulfur and thiosulfate oxidation. Investigation of metatranscriptomic data suggested that phage-directed sulfur oxidation activities showed significant increases with the increased substrate supplies in hydrothermal ecosystems, which indicates rapid and sensitive responses of virocells to altered environmental conditions. Overall, our study provides key insights on the distribution, diversity, and ecology of phage-directed dissimilatory sulfur and thiosulfate metabolisms and reinforces the need to incorporate viral contributions into assessments of biogeochemical cycling.

## Results

**Unique sulfur phages encode AMGs for oxidation of elemental sulfur and thiosulfate**. We queried the Integrated Microbial Genomes/Viruses (IMG/VR v2.1) database for phages encoding genes associated with pathways for dissimilatory sulfur oxidation and reduction processes. We identified 190 metagenomic viral contigs (mVCs) and one viral single-amplified genome[31] carrying genes encoding for reverse dissimilatory sulfite reductase subunits A and C (dsrA and dsrC), thiouridine synthase subunit E (tusE, a homolog of dsrC), sulfane dehydrogenase subunits C and D (soxC, soxD), and fused sulfur carrier proteins Y and Z for thiosulfate oxidation (soxYZ). All mVCs except one (Kilo-Moana_10000689) were estimated to be partial genome scaffolds. While phages carrying dsrA, dsrC/tusE and soxYZ have been previously described in specific marine environments, this is the first report of soxC and soxD encoded on viral genomes. Each identified mVC encoded between one to four total DSM AMGs for a total of 227 AMGs (Fig. 1a and Supplementary Data 1). The mVCs ranged in length from 5 to 308 kb, with an average length of ~31 kb and a total of 83 sequences greater than 20 kb. The mVCs consisted of 124 low-, 26 medium-, and 41 high-quality draft scaffolds according to quality estimations based on gene content (Fig. 1b). Only one mVC was a complete circular genome and was identified as previously described[22]. The majority of viruses in this study, with the exception of several mVCs encoding tusE-like AMGs were predicted to have an obligate lytic lifestyle on the basis of encoded proteins functions.

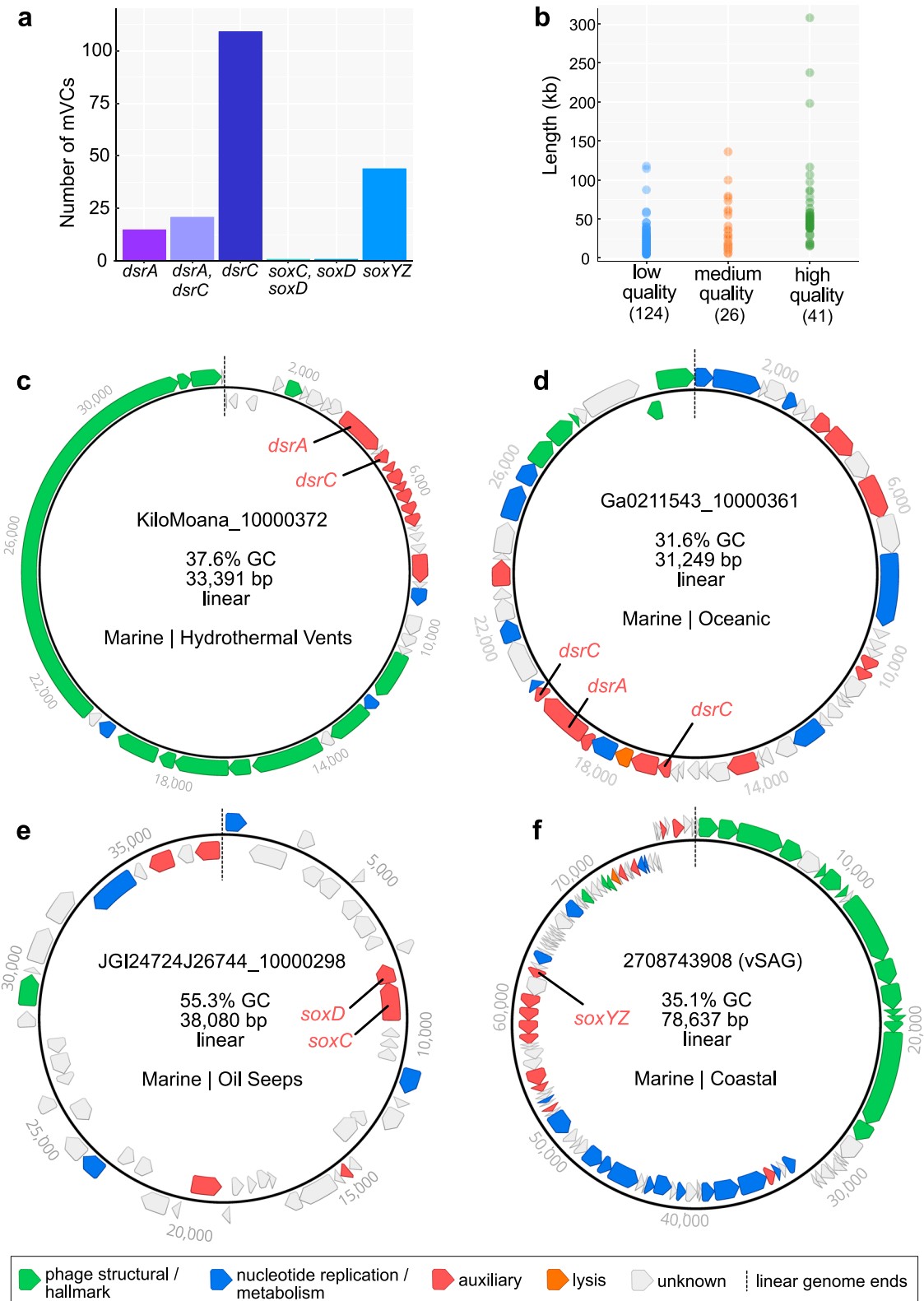

**Fig. 1 Dataset summary statistics and representative genome organization diagrams of mVCs. a** The number of mVCs, 191 total, encoding single or multiple DSM AMGs. **b** Estimated mVC genome qualities as a function of scaffold lengths. mVCs encoding **c** *dsrA* and *dsrC*, **d** *dsrA* and two *dsrC*, **e** *soxC* and *soxD*, and **f** *soxYZ*. For **c**, **d**, **e**, and **f** linear mVC scaffolds are visualized as circular with the endpoints indicated by dashed lines, and predicted open reading frames are colored according to VIBRANT annotation functions. vSAG: viral single-amplified genome, GC: guanine-cytosine content, bp: base pairs.

The mVCs displayed unique and diverse genomic arrangements, regardless of the encoded AMG(s). However, in most cases the encoded AMGs were found within auxiliary gene cassettes, separate from structural and nucleotide metabolism cassettes (Fig. 1c, d, e, f). Auxiliary cassettes in phages typically encode genes that are not essential for productive propagation but can provide selective advantages during infection, such as in specific nutrient limiting conditions or to overcome metabolic bottlenecks[32]. This genomic arrangement suggests that the role of DSM AMGs is related to host modulation rather than essential tasks such as transcription/translation, genome replication or structural assembly.

**Validation of conserved amino acid residues and domains in AMG proteins**. Validating AMG protein sequences ensures that their identification on mVC genomes represents accurate annotations (i.e., predicted biological function). We used in silico approaches for protein validation by aligning AMG protein sequences with biochemically validated reference sequences from isolate bacteria or phages and assessed the presence or absence of functional domains and conserved amino acid residues. We highlighted cofactor coordination/active sites, cytochrome c motifs, substrate-binding motifs, siroheme-binding sites, cysteine motifs, and other strictly conserved residues (collectively termed *residues*). Finally, we assessed if phage AMGs are under selection pressures to be retained.

Conserved residues identified on AMG protein sequences include: DsrA: substrate binding (R, KxKxK, R, HeR) and siroheme binding (CxgxxxC, CxxdC) (Supplementary Fig. 1); DsrC: strictly conserved cysteine motifs (CxxxgxpxpxxC) (Supplementary Fig. 2); SoxYZ: substrate-binding cysteine (ggCs) and variable cysteine motif (CC) (Supplementary Fig. 3); SoxC: cofactor coordination/active sites (XxH, D, R, XxK) (Supplementary Fig. 4); SoxD: cytochrome c motifs (CxxCHG, CMxxC) (Supplementary Fig. 5). The identification of these residues on the majority of AMG protein sequences suggests they are as a whole functional. However, there are several instances of AMGs potentially encoding non-functional or distinctively different genes. For example, only 23 DsrC AMG protein sequences contained both of the strictly conserved cysteine motifs, 112 contained only the second cysteine motif, 1 contained only the first cysteine motif, and another 5 contained neither. The lack of strictly conserved cysteine motifs in phage DsrC has been hypothesized to represent AMGs with alternate functions during infection[16], but this hypothesis has yet to be validated. Likely, most DsrC AMG protein sequences lacking one or more cysteine residues functionally serve as TusE, a related sulfur transfer protein for tRNA thiol modifications[33]. Indeed, several mVCs originating from the human oral microbiome encode *tusE*-like AMGs that flank additional *tus* genes (Supplementary Fig. 2 and Supplementary Data 2). Further examples of missing residues include two mVCs encoding *soxD* in which one is missing the first cytochrome c motif, and both are missing the second cytochrome c motif (Supplementary Fig. 5). This initially suggests the presence of non-functional SoxD, but this notion is contested by the presence of conserved residues in SoxC. Functional SoxC, encoded adjacent to *soxD* in one of the mVCs, suggests that both likely retain function. It has been shown that phage proteins divergent from respective bacterial homologs can retain their original anticipated activity or provide additional functions[34]. Overall, with the notable exception of 118 *tusE*-like AMGs, in silico analyses of AMG protein sequences suggests mVCs encode functional metabolic proteins.

To understand selective pressures on AMGs, we calculated the ratio of non-synonymous to synonymous nucleotide differences (d$N$/d$S$) in phage AMGs and their bacterial homologs to assess if phage genes are under purifying (stabilizing) selection. A calculated d$N$/d$S$ ratio below 1 indicates a gene, or genome as a whole, is under selective pressures to remove deleterious mutations. Therefore, d$N$/d$S$ calculation of mVC AMGs resulting in values below 1 would indicate that the viruses selectively retain the AMG's function by eliminating deleterious mutations in favor of those that provide function. Calculation of d$N$/d$S$ for mVC *dsrA*, *dsrC* and *soxYZ* AMGs resulted in values below 1, suggesting AMGs are under purifying selection (Supplementary Fig. 6).

**DSM AMGs likely manipulate key steps in sulfur oxidation pathways to redistribute energy**. As previously stated, DSM AMGs encoded by the mVCs likely function specifically for the manipulation of sulfur transformations in the host cell during infection. To better understand the implications of this manipulation, we constructed conceptual diagrams of both sulfur (i.e., *dsr* AMGs) oxidation and thiosulfate (i.e., *sox* AMGs) oxidation/disproportionation in both uninfected and infected hosts (Fig. 2).

To understand the potential advantages of carrying *dsrC* and *dsrA* AMGs specifically, each step in the sulfide oxidation pathway needs consideration. During host-only sulfide oxidation[35], sulfide diffusing into the cell is converted into elemental sulfur by a sulfide: quinone oxidoreductase (e.g., *sqr*) and in some cases the pathway can begin directly with the import of elemental sulfur. The elemental sulfur can be stored in localized sulfur globules until it is metabolized through the sulfide oxidation pathway[36]. During sulfide oxidation, elemental sulfur carried by the sulfur carrier protein DsrC is oxidized into sulfite by the enzyme complex DsrAB. This step is estimated to be the rate limiting step in the complete pathway and yields the most electrons (six electrons) for ATP generation. Rate limitation is caused by either the saturation of the DsrAB enzyme complex or the DsrC carrier[37,38]. The final steps in sulfide/sulfur oxidation involve further oxidation of sulfite into adenosine 5-phosphosulfate (APS) and then sulfate by an APS reductase (e.g., *aprAB*) and sulfate adenylyltransferase, respectively (e.g., *sat*), which yields two electrons[35]. The obtained ATP can then be utilized for cellular processes. In contrast, during phage infection involving the modulation of sulfide oxidation, the rate limiting step (i.e., co-activity of DsrC and DsrA) can be supplemented by phage DsrC and/or DsrA to potentially increase the rate and ATP yield of the reaction, as well as utilize any stored elemental sulfur[22]. This influx of ATP could then be effectively utilized for phage propagation (e.g., phage protein production, genome replication, or genome encapsidation) (Fig. 2a).

Likewise, the normal state of thiosulfate oxidation/disproportionation may be augmented by phages encoding *soxYZ*, *soxC* and *soxD*. During host-only thiosulfate oxidation[39], thiosulfate is transported into the cell where the two thiol groups, transported by SoxYZ, undergo a series of oxidation reactions. A portion of the carried sulfur, after yielding two electrons, will be transported out of the cell as sulfate. The remaining carried sulfur may either be stored in elemental sulfur globules or proceed to the key energy yielding step. The key energy yielding step bypasses the storage of elemental sulfur and utilizes the SoxCD enzyme complex to produce six electrons for ATP yield[35,40]. During phage infection involving the modulation of thiosulfate oxidation/disproportionation, the entire pathway can be supported by both host and phage SoxYZ sulfur carriers in order to continuously drive elemental sulfur storage, which could then be oxidized by the Dsr complex. However, there is no evidence that phages benefit from coupling the *sox* and *dsr* pathways since no mVCs were found to encode both a *sox* and *dsr* AMG simultaneously. Finally, phage SoxCD may be utilized to drive the

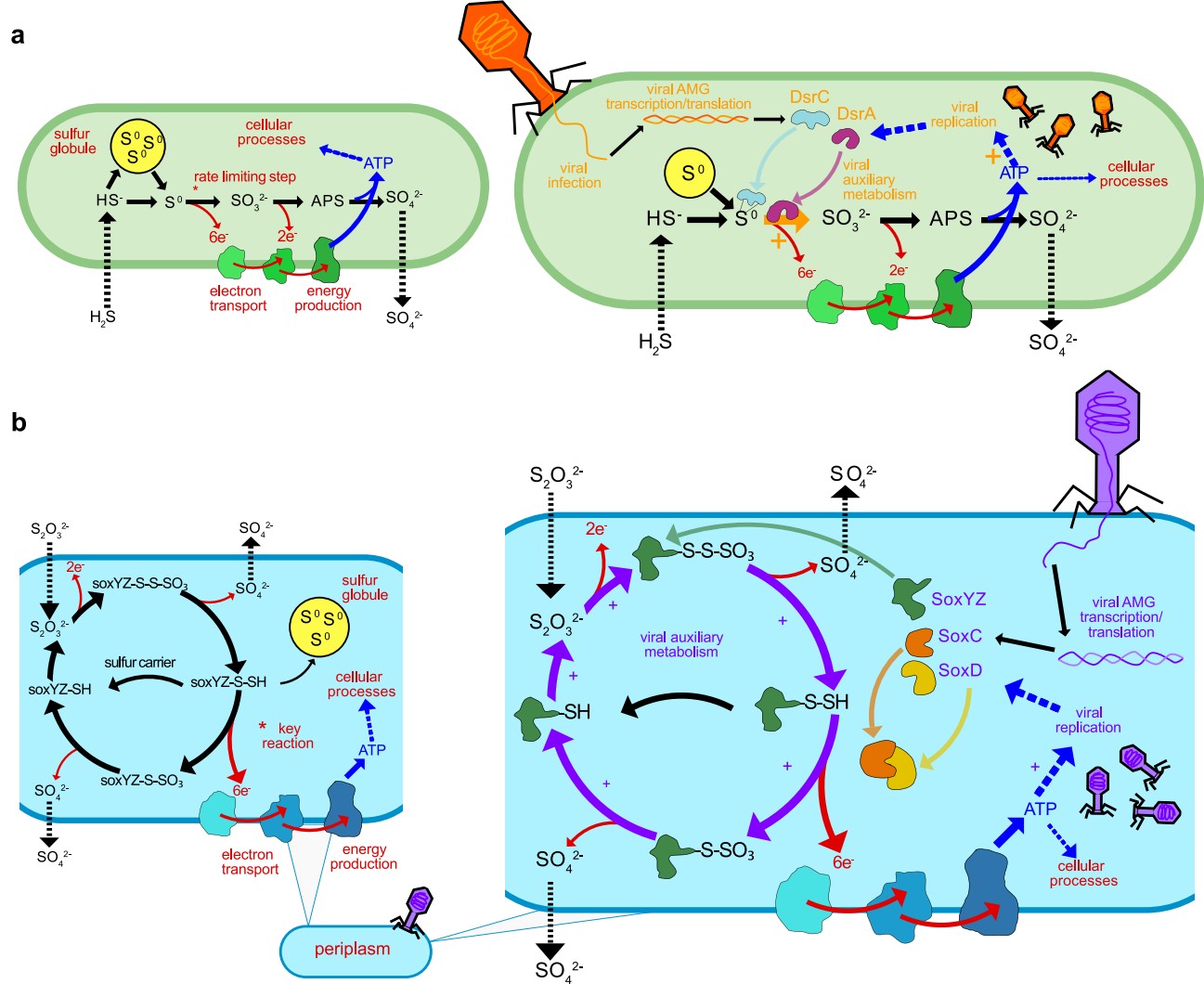

**Fig. 2 Conceptual diagrams of viral DsrA, DsrC, SoxC, SoxD, and SoxYZ auxiliary metabolism. a** Microbial dissimilatory oxidation of hydrogen sulfide and stored inorganic sulfur. The resulting production of ATP utilized for cellular processes and growth and the pathway's rate limiting step is indicated with an asterisk (left). Viral infection and manipulation of sulfur oxidation by encoded DsrA or DsrC to augment the pathway's rate limiting step and increase energy yield towards viral replication (right). **b** Microbial dissimilatory oxidation of thiosulfate or storage of inorganic sulfur in the periplasm. The resulting production of ATP is utilized for cellular processes and the pathway's key energy yielding reaction indicated with an asterisk (left). Viral infection and manipulation of thiosulfate oxidation by encoded SoxC, SoxD, or SoxYZ to augment the entire pathway and the key energy yielding step to increase energy yield towards viral replication (right). For **a** and **b** cellular processes are shown in red, sulfur oxidation pathway is shown in black, energy flow is shown in blue, and viral processes are shown in orange (**a**) or purple (**b**). For all pathway steps shown, microbial enzymes and sulfur carriers are functional in tandem with viral augmentation. APS: adenosine 5′-phosphosulfate.

pathway to the key energy yielding step. As with the *dsr* pathway, the resulting ATP would be utilized for phage propagation (Fig. 2b).

**Sulfur phages are widely distributed in the environment**. Next, we studied the ecological and distribution patterns of mVCs encoding DSM AMGs. We characterized their diverse ecology and distribution patterns in various environments by building phylogenetic trees using the identified AMG and reference microbial proteins, and parsing environmental information of mVC metadata from the IMG/VR database. We identified mVCs encoding *dsrA* mainly in a few ocean environments, while more widely distributed mVCs encoding *dsrC* were found in in ocean, saline, oil seep-associated, terrestrial, engineered, and symbiotic environments (Fig. 3a, b). For *soxC* and *soxD*, we only identified mVCs encoding these AMGs in two metagenome datasets, one

from Santa Barbara Channel oil seeps (mVC encoding both *soxC* and *soxD*) and another from freshwater sediment from Lake Washington (Fig. 3c, d). The mVCs encoding *soxYZ* were discovered in aquatic environments, consisting of different ocean, saline and freshwater ecosystem types (Fig. 3e). In addition to mVC distribution among diverse ecosystem types we identified wide biogeographic distribution across the globe (Fig. 3f). Collectively, these DSM AMGs are ecologically and biogeographically ubiquitous, and potentially assist host functions in many different environment types and nutrient conditions (including both natural and engineered environments).

**Sulfur phages are taxonomically diverse within the order *Caudovirales***. We applied two approaches to taxonomically classify and cluster the identified mVCs. First, we used a reference database similarity search to assign each mVC to one of 25

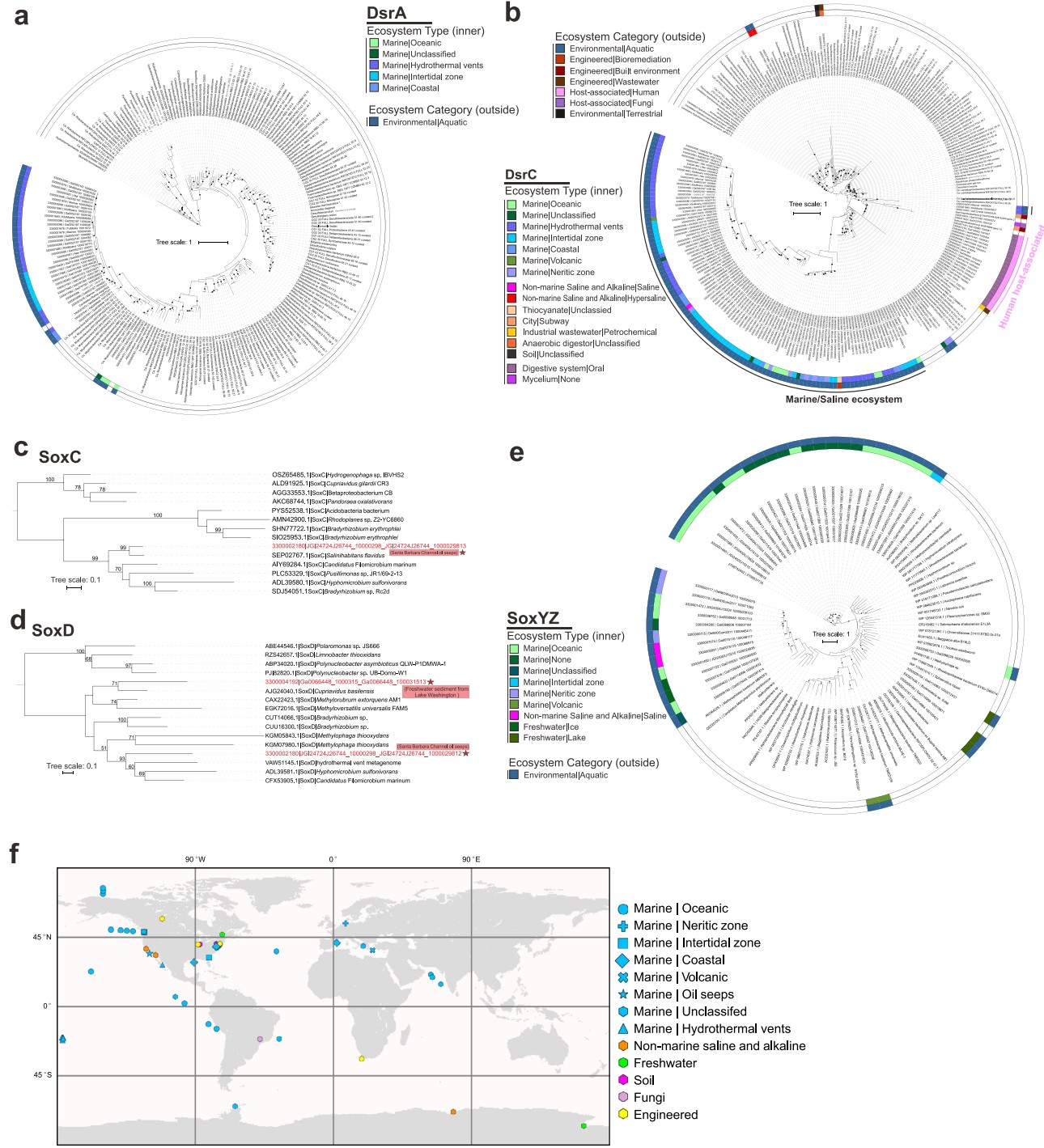

**Fig. 3 Phylogenetic tree of AMG proteins and distribution of phage genomes (on a world map). a**, **b** Phylogenetic trees of phage DsrA and DsrC. **c**, **d**, **e** SoxC, SoxD, SoxYZ. Ultrafast bootstrap (UFBoot) support values (>50%) are labeled on the nodes. **c**, **d** Phage gene encoded protein sequences are labeled with stars and their environmental origin information is labeled accordingly. The ecosystem type (inner ring) and ecosystem category (outside ring) are provided for phage genomes in the phylogenetic trees in **a**, **b**, **e**; different colors represent different ecosystem type and ecosystem category in the legends, blank places in each ring are for microbial references. **f** World map showing distribution of phage genomes that contain the sulfur-related AMGs. Studies on human systems are excluded from the map. Different ecosystem types are represented by different symbols and colors in the legend.

different prokaryote-infecting viral families (see Methods). The majority of mVCs were assigned to *Myoviridae* (132 mVCs; 69%), *Siphoviridae* (43 mVCs; 22%) and *Podoviridae* (9 mVCs; 5%). These three families represent dsDNA phages belonging to the order *Caudovirales*. The remaining seven mVCs were identified as ambiguous *Caudovirales* (3 mVCs; 1.5%) and unknown at both the order and family levels (4 mVCs; 2%). However, based on the

data presented here and previous classifications[16,22,30], the seven unclassified mVCs likely belong to one of the three major *Caudovirales* families (Fig. 4).

In accordance with these results we constructed a protein sharing network of the mVCs with reference viruses from the NCBI GenBank database (Fig. 4). The mVCs arranged into four main clusters with reference *Myoviridae*, *Siphoviridae* and

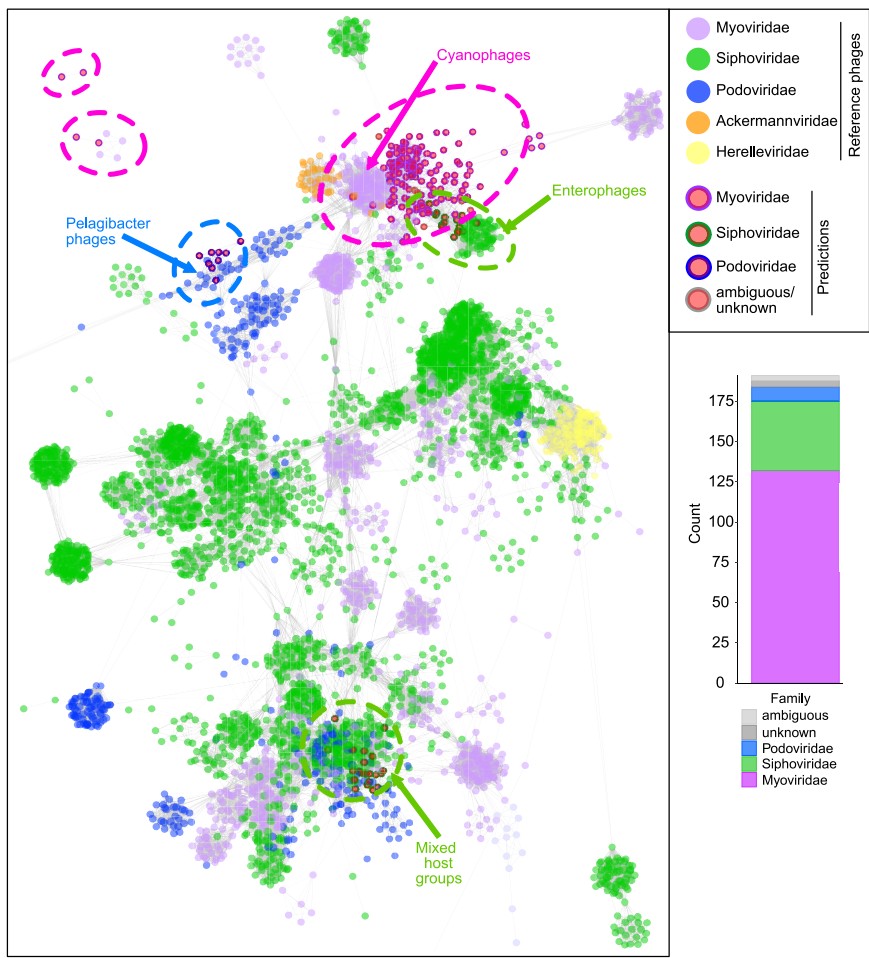

**Fig. 4 Taxonomic assignment of mVCs and protein network clustering with reference phages.** In the protein network each dot represents a single mVC (circles with outlines) or reference phage (circles without outlines), and dots are connected by lines respective to shared protein content. Genomes (i.e., dots) having more similarities will be visualized by closer proximity and more connections. Cluster annotations depicted by dotted lines were approximated manually. mVC taxonomy was colored according to predictions by a custom reference database and script, shown by bar chart insert.

*Podoviridae*, and four individual mVCs were arranged outside of main clusters. Of the seven mVCs with ambiguous/unknown predictions, six clustered with *Myoviridae* and *Siphoviridae* mVCs and reference phages, further suggesting their affiliation with major *Caudovirales* families. Overall, the network diagram validated the reference-based taxonomic assignment results (i.e., mVCs predicted to be podoviruses clustered with reference podoviruses, as with myoviruses and siphoviruses). On the basis of these findings, we hypothesize that the function(s) of DSM AMGs during infection is most likely constrained by specific host sulfur metabolisms rather than viral taxonomy. The broad distribution of DSM AMGs across *Caudovirales* further suggests that this modulatory mechanism is established across multiple taxonomic clades of phages, either arising independently or acquired via gene transfer. Most mVCs clustered with reference phage genomes of varying taxonomy and host ranges, though there was not significant enough protein similarity between the mVCs and these reference phages to suggest similarity at the genus level. It is likely that host range stems beyond these indicated taxa, suggested by the inclusion of a SUP05-infecting mVC[22] within the *Pelagibacter* cluster. In the present state of the reference databases, this type of protein sharing network cannot be used to reliably predict the host range of these uncultivated mVCs, but rather is indicative of shared taxonomy (e.g., mVC podoviruses clustering with reference podoviruses, and likewise for myoviruses and siphoviruses). Based on phylogeny and AMG

protein similarity, the mVC host range appears to be primarily Gammaproteobacteria from the SUP05/*Thioglobus* clades, with the possibility of extended host range to *Methylophilaceae* in the Betaproteobacteria (Fig. 3). Using CRISPR analysis against 7178 spacers from 25 metagenomes we were unable to validate any mVC link to a putative host.

**Sulfur phages display diversification across environments and genetic mosaicism.** To further assess the diversity of the identified mVCs and their evolutionary history, we analyzed shared protein groups as well as gene arrangements between individual mVCs. All predicted proteins from 94 of the mVCs, excluding mVCs encoding only *tusE*-like AMGs, were clustered into protein groups. Our protein clustering method for featuring the diversity of the mVCs, despite representing partial genome sequences, was assessed and verified using *Caudovirales* phages from NCBI RefSeq (see Methods). A total of 794 protein groups representing 3677 proteins were generated, roughly corresponding to individual protein families. Only a few protein groups were globally shared among the mVCs, including common phage proteins (e.g., *phoH*, *nifU*, *iscA*, nucleases, helicases, lysins, RNA/DNA polymerase subunits, ssDNA-binding proteins and morphology-specific structural proteins) (Fig. 5a). A lack of shared protein groups between the mVCs may be anticipated due to missing genes on the partial mVC scaffolds. However, distinct phage

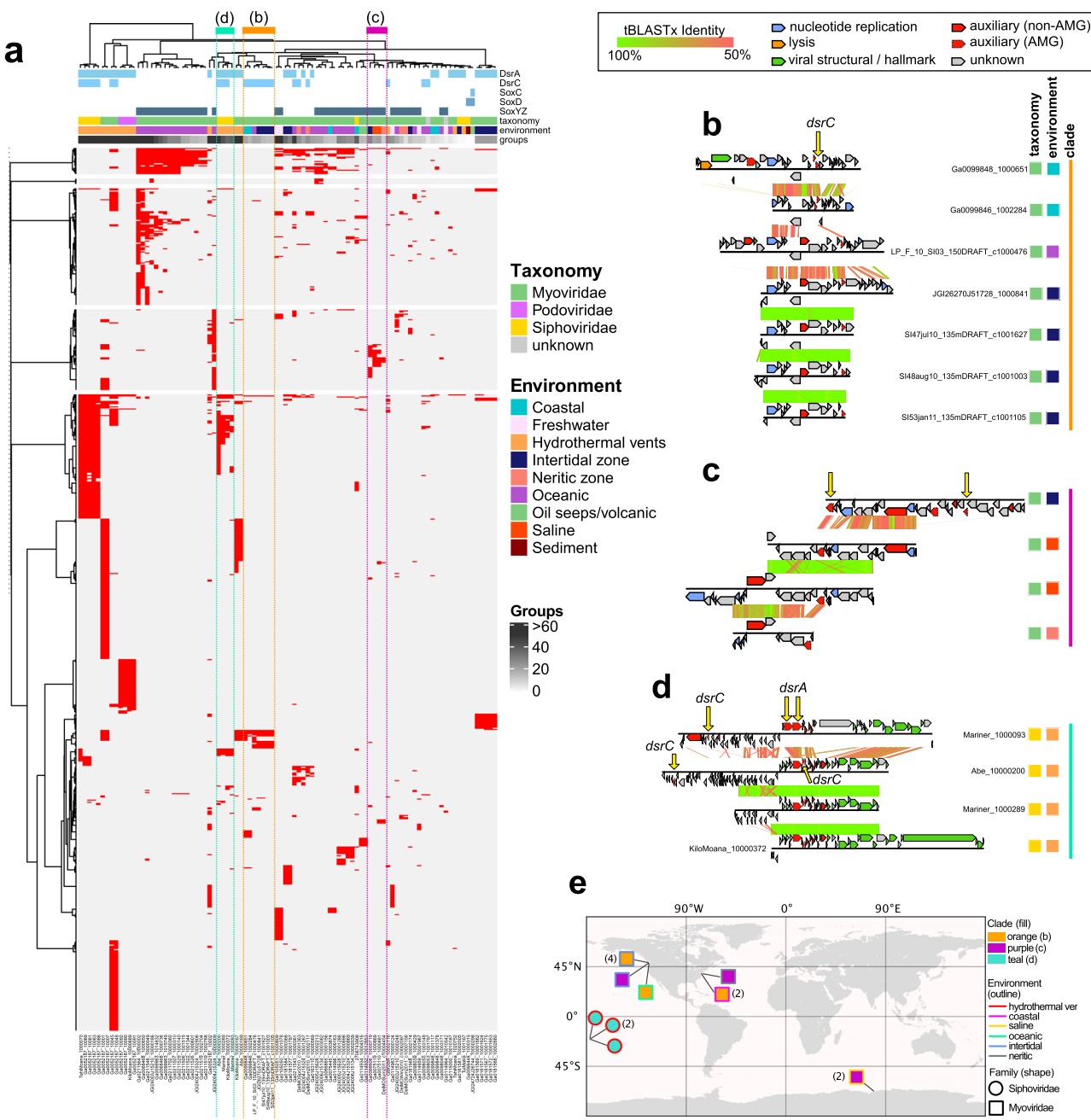

**Fig. 5 mVC protein grouping and genome alignments. a** mVC hierarchical protein grouping where each row represents a single protein group (887 total) and each column represents a single mVC (94 total). Metadata for encoded AMGs, estimated taxonomy, source environment and number of protein groups per mVC is shown. Clades respective of **b**, **c**, and **d** are depicted by colored dotted lines. Genome alignments of **b** seven divergent *Myoviridae* mVCs encoding *dsrC* from diverse environments, **c** four divergent *Myoviridae* mVCs encoding *soxYZ* from diverse environments, and **d** four divergent *Siphoviridae* mVCs encoding *dsrA* and *dsrC* from hydrothermal environments. For the genome alignments, each black line represents a single genome and arrows represent predicted proteins, which are colored according to VIBRANT annotations; genomes are connected by lines representing tBLASTx similarity. **e** Map of geographic distribution of 15 mVCs depicted in **b**, **c**, and **d** annotated with respective clade, source environment, and taxonomic family.

lineages share few protein groups regardless of genome completeness. Overall, the results of the protein grouping are consistent with that of taxonomic clustering, further highlighting the diversity of phage genomes that encode DSM AMGs. A lack of universally shared protein groups likewise suggests the DSM AMGs function independently of other host metabolic pathways and likely strictly serve to supplement host DSM pathways.

To identify if the shared protein groups are relevant to the DSM AMGs and further highlight mVC diversity we generated a second set of protein clusters corresponding to five proteins before and five proteins after the DSM AMG, including the AMG. Since the true completeness of the entire mVC cannot be determined, this subset of 11 proteins adjacent to the DSM AMGs was utilized to best represent potential shared features regardless of completion. This second set included 70 mVCs (we excluded 24 mVCs for which the encoded DSM AMG was within five genes of a scaffold end). In total, 116 protein clusters were generated (Supplemental Fig. 7). Interestingly, nearly identical proteins were common, namely PhoH, NifU, IscA, GrxD, TusA, NrdAB, RNA/DNA polymerase subunits, ssDNA-binding

proteins and morphology-specific structural proteins. However, the shared groups represented only a small subset of all groups. Therefore, in addition to the common functions such as iron-sulfur cluster formation (e.g., NifU, IscA, GrxD and TusA), the mVCs encode dissimilar proteins, likely resulting from varied evolutionary backgrounds.

Most mVCs that formed clades according to whole mVC shared protein groups could be explained by shared taxonomy and/or source environment. This observation further validates that despite the mVCs representing partial scaffolds, they encoded sufficient information to be accurately grouped into clades. That is, similar mVCs by genome alignment, as with taxonomy and source environment, were found to group into the same clade. For example, 16 *Myoviridae* mVCs encoding *soxYZ* from oceanic environments clustered together, only differing according to their total number of representative protein groups (Fig. 5a). There were exceptions, such as seven *dsrC*-encoding mVCs, which displayed variable pairwise protein similarity (at a 50% identity cutoff) and variation in the location of their *dsrC* gene within their genome, despite a clearly shared and distinctive synteny of other genes (Fig. 5b). The seven mVCs originated from three different marine environment types (coastal, oceanic and intertidal) and were all predicted to be myoviruses (Fig. 5b). This diversity is likely explained by the retention of the *dsrC* gene over time despite components of the genome undergoing genetic exchange, recombination events or mutation accumulation. Phages are well known to display genetic mosaicism, or the exchange and diversification of genes and gene regions[32,41]. The same conclusion can be made with myoviruses encoding *soxYZ* from different marine environments (intertidal, saline and neritic) (Fig. 5c), as well as siphoviruses encoding both *dsrC* and *dsrA* from hydrothermal environments (Fig. 5d). In addition to distribution among diverse environmental categories these genetically mosaic mVCs, per protein sharing clade, are geographically dispersed (Fig. 5e). Additionally, one mVC (Ga0066606_10000719) encoding *soxYZ* also encodes the assimilatory sulfur metabolism AMG *cysC* (Fig. 5b). This presents an interesting discontinuity suggesting that this particular mVC, as well as three others encoding *cysC* (Ga0052187_10001, Ga0052187_10007 and JGI24004J15324_10000009), target both dissimilatory and assimilatory sulfur metabolism simultaneously to more generally affect sulfur metabolism in the host.

**Estimations of sulfur phage contributions to sulfur oxidation based on omics-data analyses.** We utilized metagenomic datasets containing the mVCs to calculate the ratio of phage:total genes for each AMG. The phage:total gene ratios within a community and for each predicted phage-host pair can be used to estimate phage contributions to sulfur and thiosulfate oxidation/disproportionation. This relies on the assumption that gene ratios can proportionally reflect real metabolic activities, and that host cells in a virocell state retain the same level of environmental fitness as compared to uninfected microorganisms. By mapping metagenomic reads to AMGs and putative bacterial hosts within the metagenome, we obtained the mVC AMG to total gene ratios, which represents the relative contribution of AMG functions to the representative metabolism such as sulfur oxidation (Supplementary Data 3, 4 and Supplementary Fig. 8). We calculated mVC *dsrA* (Fig. 6a) and *soxYZ* (Fig. 6b) gene coverage ratios in hydrothermal, freshwater lake, and *Tara* Ocean metagenomic datasets. We identified phage-host gene pairs that contained mVC AMGs and their corresponding host genes from the phylogenetic tree of DsrA and SoxYZ (Supplementary Figs. 9 and 10). Our results show that phage *dsrA* contributions in hydrothermal environments arise primarily from the SUP05

Gammaproteobacteria Clade 2; and those of phage *soxYZ* are niche-specific, with Lake Croche, Lake Fryxell, and *Tara* Ocean samples mainly represented by Betaproteobacteria Clade, *Methylophilales*-like Clade, and Gammaproteobacteria Clade, respectively. This indicates the specificity of AMGs being distributed and potentially functioning in each environment. The average phage:total gene coverage ratios also differ in individual groups, with phage *soxYZ*:total ratio in *Tara* Ocean samples being the highest (34%), followed by phage *dsrA*:total ratio in hydrothermal samples (7%) and phage *soxYZ*:total ratio in freshwater lakes (3%). Phage *soxYZ*, the sulfur carrier gene, in the oceans have higher phage:total gene coverage ratio compared to *dsrA*, a component of the catalytic core of Dsr complex, in the other two environments. *Tara* Ocean samples used here are all from epipelagic zones characterized as oxygenated layers with low concentrations of sulfur[42], while plume samples (Lau Basin) are from deep-ocean hydrothermal ecosystems with high concentrations of sulfur[43]. Nevertheless, along with observations associated with phage *dsrC*, our results suggest that AMGs encoding sulfur carriers rather than catalytic subunits appear to be more favored by phages. These findings were unexpected since we expected epipelagic environments to have lower sulfur oxidation activity in comparison to hydrothermal plumes. While the limited environment types, conditions, and sulfur AMGs studied here do not provide sufficient statistical confidence to generalize these results, especially when comparing different genes from separate environments, higher abundance of sulfur carrier genes in phage nevertheless could still be a common phenomenon. Additionally, although gene abundance ratios do not necessarily represent function contributions, this scenario still provides a reasonable estimation to suggest considerable sulfur-oxidizing contributions of phage sulfur AMGs in virocells.

Subsequently, the phage:host AMG coverage ratios for individual phage-host pairs were calculated to estimate the potential functional contribution within each environmental sample (Fig. 7a, b; Supplementary Data 3, 4 and Supplementary Figs. 11 and 12). By taking average ratios of groups of *dsrA* phage-host pairs in SUP05 Clade 1 and SUP05 Clade 2, and *soxYZ* phage-host pair in freshwater lake and *Tara* Ocean samples, we found that within each pair the phage:total gene coverage ratios were generally higher than ~50%. These within-pair phage:total gene coverage ratios are much higher than the above phage:total ratios in the whole community. *Tara* Ocean samples also have the highest average phage:total gene coverage ratios of phage-host pairs among these three environments, as with the pattern of ratios in the whole community. To estimate the percentage of virocells in the community, we use average values of 16% and 15% for marine (range of 3–31% or 3–26% in free-living and particulate-associated marine bacteria)[10,44] and freshwater lake (range of 1 to 17%+/−12%)[45] bacteria. The estimated phage:total gene coverage ratio within the whole community should be the virocell percentage multiplied by the average phage:total gene coverage ratio within phage-host pairs (as the phage gene coverage), and then divided by total gene coverage. We found that estimated ratios are not consistent with the observed ratios (*Tara* Ocean: estimated ratio 68% versus the observed ratio of 34%; hydrothermal environment: estimated ratio of 15–38% versus observed ratio of 12–20%; freshwater lake: estimated ratio of 27% versus observed ratio of 3.1%). This could result from an unknown fraction of the host cells being infected by phages that do not contain DSM AMGs as these virocells do not contribute phage genes to sulfur metabolism and/or the percentages of cells in a virocell state being below the average levels.

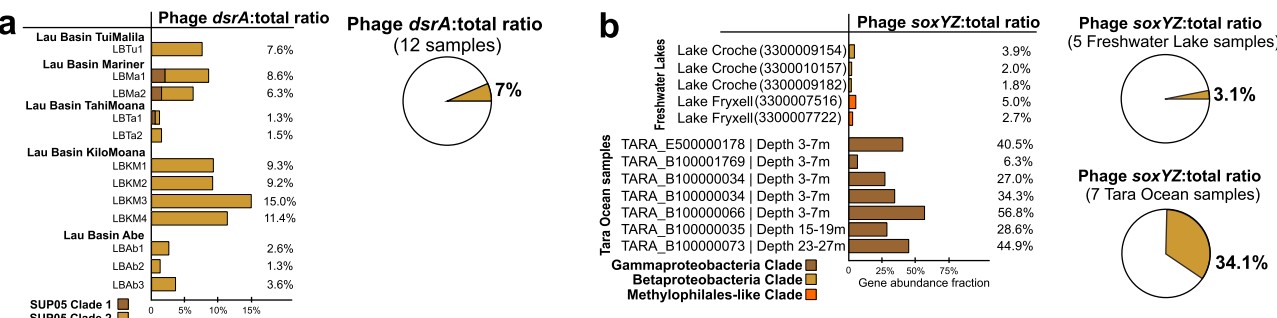

**Fig. 6 Phage to total *dsrA* and *soxYZ* gene coverage ratios. a** Phage *dsrA* to total (phage and bacterial *dsrA* gene together) gene coverage ratios. The contribution of phage *dsrA* genes from different SUP05 Gammaproteobacteria clades is shown in different colors. The average phage *dsrA*:total ratio was calculated from 12 samples. **b** Phage *soxYZ* to total gene coverage ratios. The contribution of phage *soxYZ* genes from three different clades is shown in different colors. Genes from freshwater lake and *Tara* Ocean samples were compared separately, and the average phage *soxYZ*:total ratios were calculated and compared separately as for freshwater lake and *Tara* Ocean samples. *Tara* Ocean sample IDs were labeled and the corresponding metagenome IDs were listed in Supplementary Data 4. LBTu: Lau Basin Tui Malila, LBMa: Lau Basin Mariner, LBTa: Lau Basin Tahi Moana, LBKM: Lau Basin Kilo Moana, LBAb: Lau Basin Abe.

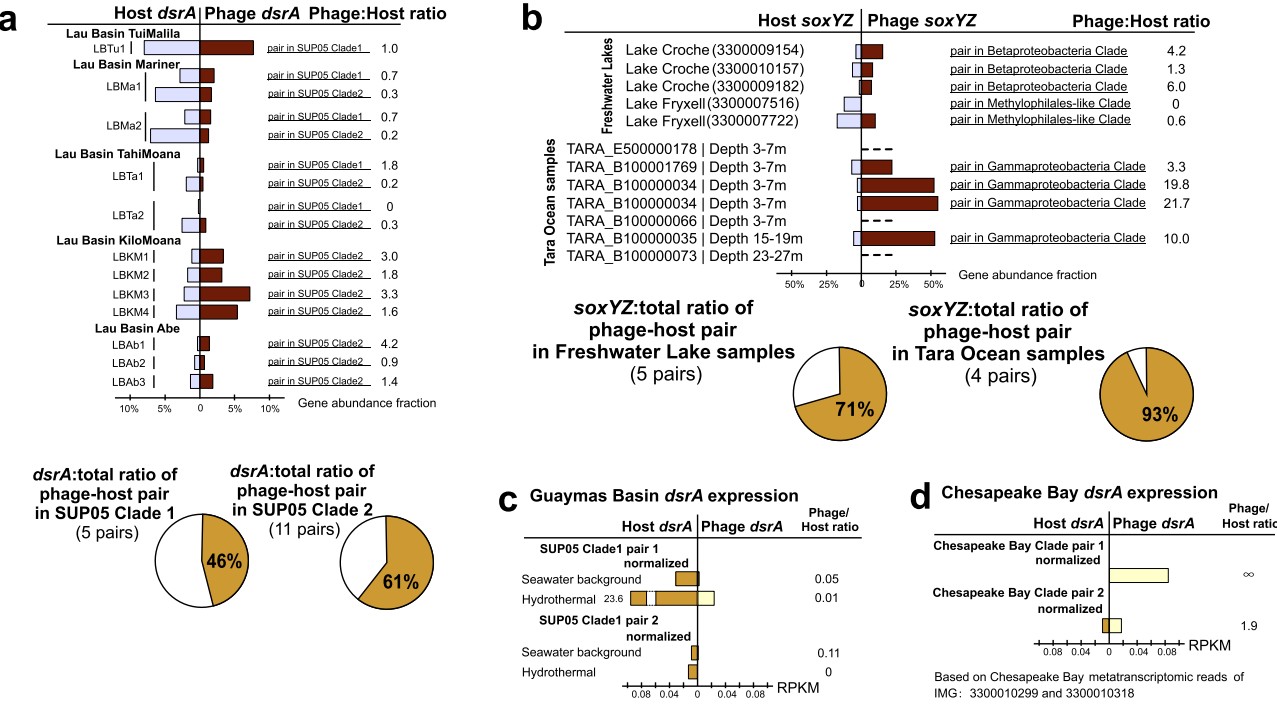

**Fig. 7 Phage to host *dsrA* and *SoxYZ* gene coverage ratios and *dsrA* gene expression comparison between phage and host pairs. a** Phage *dsrA* to total gene coverage ratios of each phage-host pair. Average phage *dsrA*:total ratios of phage-host pairs in SUP05 Clade 1 and Clade 2 were calculated by 5 and 11 pairs of genes, respectively. **b** Phage *soxYZ* to total gene coverage ratios of each phage-host pair. The contribution of phage *soxYZ* genes from three different clades is shown in different colors. Average phage *dsrA*:total ratios of phage-host pairs in freshwater lakes and *Tara* Ocean were calculated separately. *Tara* Ocean sample IDs were labeled and the corresponding metagenome IDs were listed in Supplementary Data 4. **c** Phage to host *dsrA* gene expression comparison in Guaymas Basin metatranscriptomes. The same database was used for mapping both hydrothermal and background metatranscriptomic datasets. **d** Phage to host *dsrA* gene expression comparison in Chesapeake Bay metatranscriptomes. The same database was used for mapping all Chesapeake Bay metatranscriptomic datasets. Gene expression levels are shown in RPKM normalized by gene sequence depth and gene length. LBTu: Lau Basin Tui Malila, LBMa: Lau Basin Mariner, LBTa: Lau Basin Tahi Moana, LBKM: Lau Basin Kilo Moana, LBAb: Lau Basin Abe, RPKM: reads per kilobase per million mapped reads.

The above analyses suggest that DSM AMGs likely contribute significantly to function of host-driven metabolisms on the scale of both community level and individual phage-host pairs, while the ratio of contribution varies greatly for each environment and each niche-specific AMG. Importantly, phage-encoded *soxYZ* have a high gene coverage contribution to pelagic ocean microbial communities, which highlights the functional significance of phage-driven sulfur cycling metabolisms, and that of thiosulfate

oxidation/disproportionation as a whole in this environment, which remains critically under-studied[16,46].

**Rapid alteration of sulfur phage *dsrA* activity across geochemical gradients.** Since DSM AMGs are associated with critical energy generating metabolism in microorganisms, we wanted to study the ability of sulfur phages to respond to changing geochemistry, involving virocell-driven biogeochemical cycling. In

hydrothermal ecosystems, reduced chemical substrates such as $H_2S$, $S^0$, $CH_4$, and $H_2$ display sharp chemical gradients as they are released from high-temperature vents and dilute rapidly upon mixing with cold seawater. Microorganisms in deep-sea environments respond to such elevated concentrations of reduced sulfur compounds by upregulating their metabolic activity in hydrothermal environments[43,47]. These characteristics make hydrothermal and background deep-sea environments a contrasting pair of ecological niches to investigate alteration of AMG expression. We used transcriptomic profiling to study gene expression in phage:host pairs recovered from hydrothermal vents in Guaymas Basin and background deep-sea samples in the Gulf of California (Supplementary Data 3 and Supplementary Figs. 11 and 12). Sulfur phage *dsrA* expression measured in reads per kilobase per million mapped reads (RPKM) varied from 0.03 to 3 in the background deep-sea to 0.40 to 39 in hydrothermal environments (Supplementary Data 3). Average phage *dsrA* expression ratio of hydrothermal to background was 15 (Supplementary Data 3). Limited by coding gene repertoire and their biology, phages themselves do not have the ability to independently sense and react to sulfur compounds. However, our results suggest that sulfur phage activities, occurring within a virocell, are closely coupled to changing geochemistry with higher observed activity in environments with greater concentration of reduced sulfur compounds.

In Guaymas Basin hydrothermal environments, as reflected by two pairs of SUP05 Clade 1 phage and host *dsrA* genes, phage to host *dsrA* transcript ratios varied from 0 to 0.11 (Fig. 7c). In contrast, in Chesapeake Bay, as reflected by two pairs of phage and host *dsrA* transcripts (Chesapeake Bay *dsrA* clade), phage to host *dsrA* transcript ratios varied from 1.9 to infinity (host transcript abundance is zero). The low abundance of phage *dsrA* in hydrothermal metatranscriptomes is in sharp contrast to the high abundance of phage *dsrA* in hydrothermal metagenomes (observed at Guaymas Basin and Lau Basin) (Fig. 7a, c). One explanation for this observation is that this scenario could be an accident but not representative of real phage gene expression patterns in hydrothermal systems, possibly occurring in a situation when phage activity was very high just prior to sampling. In this scenario, the majority of hosts/virocells might have lysed post viral infection.

## Discussion

Since the first descriptions of viral metabolic reprogramming using AMGs[13] there has been interest in the extent and overall impact of viral auxiliary metabolism on global energy flows and ecosystem nutrient availability[48]. Through metagenomic surveys and investigation, we have expanded the current understanding of viral auxiliary metabolism impacting dissimilatory sulfur oxidation processes. Specifically, we have shown that diverse lineages of phages are involved in these processes, investigated their biogeography, ecology, and evolutionary history, and estimated their potential effects on microbiomes. From this, several hypotheses and questions regarding viral auxiliary metabolism and sulfur cycling can be addressed.

First, our findings support previous hypotheses that viral metabolism targets key or bottleneck steps in host metabolic pathways. DsrA, DsrC, SoxYZ, SoxC, and SoxD all alleviate bottlenecks in sulfur and thiosulfate oxidation/disproportionation[22,49]. We did not identify other genes in sulfur oxidation pathways such as sulfide: quinone oxidoreductase, flavocytochrome *c* cytochrome/flavoprotein subunits, APS reductase subunits, sulfate adenylyltransferase, *dsrB*, or *soxAB* for other necessary steps of sulfur oxidation. However, this poses the additional question of why DsrB, the dimer pair to DsrA, has yet to be identified as an AMG. Furthermore,

sulfur carriers, rather than enzymes, appear to be more favored by phages. In total, 174 mVCs in this study encoded at least one sulfur carrier (*dsrC*, *tusE*-like, *soxYZ*) with only the remaining 17 encoding catalytic subunits of enzymes (*dsrA*, *soxC*, *soxD*). Phage sulfur carriers were observed to be more abundant in the phage community than catalytic subunits such as *dsrA*. This may be due to the greater need for sulfur carriers (e.g., *dsrC*) to drive dissimilatory sulfur transformations. Evidence for this hypothesis is provided by observations that sulfur carriers are often constitutively expressed in host cells in comparison to respective catalytic components (e.g., *dsrA*)[38,50]. By providing transcripts and proteins of these important pathway components during infection, phages encoding DSM AMGs may benefit more from obtaining greater energy and self-catalyzing substrates within a virocell.

The data presented by mVC protein clustering and genome alignments (Fig. 5) supports the hypothesis that the DSM AMGs are retained on fast evolving phage genomes, pointing specifically to a role of the AMG in increasing phage replication abilities and fitness. Although the mechanism of dispersion is unknown for most of the mVCs, it is likely that a single AMG transfer event occurred within each clade based on retention of similar gene arrangements at AMG locations in the respective genomes. This suggests that the AMG were retained despite niche (i.e., geographic and environmental) differentiation of individual mVC populations. It has been postulated that AMGs, like other phage genes, must provide a significant fitness advantage in order to be retained over time on an evolving phage genome[12].

Taken together, these observations support the conclusion that viral auxiliary metabolism targets key steps in host metabolic pathways for finely tuned, host-dependent manipulation of energy production or nutrient acquisition. Although the fitness effects of DSM AMGs have not been quantified in a model system, the geographical distribution of identified mVCs and retention of AMGs by phages despite constrained coding capacity strongly suggests a significant fitness benefit of encoding DSM AMGs. The exact fitness benefit achieved from encoding DSM AMGs remains elusive without cultured representatives of phage-host pairs and subsequent genetic manipulation abilities. Furthermore, a model system would be beneficial for elucidating the functionality of the AMGs, beyond the evidence from protein domain analyses presented here. Although most AMGs encoded conserved functional domains and residues, the identification of divergent sequences, such as *tusE*-like AMGs encoding a single cysteine residue or *soxD* AMGs that appear to lack a cytochrome c motif, necessitates further biochemical evaluation of AMG-encoded proteins. For example, a divergent PebA encoded by a cyanophage was found to short-circuit the original host pathway by excluding the necessity of the subsequent host enzyme PebB[34]. It is possible sulfur AMG-encoded proteins likewise short-circuit or increase the rate of host sulfur oxidation pathways using divergent AMGs.

Since DSM AMGs have been identified on phages from all three major *Caudovirales* families it is likely that the fitness benefits deal specifically with sulfur oxidation and electron yield from bolstering the speed or efficiency of the pathway, rather than phage taxonomy-dependent reasons. Based on evidence from systems with cyanophages encoding photosystem AMGs, a potential utility of sulfur AMGs in bolstering the speed or efficiency of the pathway would be to increase the yield rate of dNTPs for genome replication[14]. Further evidence would suggest the AMGs could also function to upregulate the expression of important metabolic genes that encode for unstable protein products[51]. In such cases the host cell is adapted for such metabolic constraints but the replication rate of the phage is directly dependent on the translation rate of the given AMG protein product[51]. Therefore, the phage, rather than the host,

benefits from an additional copy of the metabolic gene leading to recombination and retention of the AMG on the phage genome. It is most likely that the phages benefit primarily in the short term and during active lytic infection due to the abundance of DSM AMGs on lytic phage genomes. Yet, the presence of assimilatory sulfate reduction genes (i.e., *cysC*) in conjunction with DSM genes provides an example of a possible exception with a more general sulfur manipulation, highlighting the necessity of further investigations into viral auxiliary metabolism.

The abundance of phage DSM AMGs in metagenomes and metatranscriptomes as measured by phage:total gene coverage ratios suggest that phage-mediated reduced sulfur transformations can contribute significantly to fluxes and budgets of sulfur within the community (Fig. 8 and Supplementary Figs. 8 and 13). Within each phage-host pair, phage genes contribute to over half of gene coverage associated with the sulfur and thiosulfate oxidation pathways, which highlights the underappreciated role of phages encoding DSM AMGs in remodeling sulfur cycling, especially for the oxidation of reduced sulfur. Reduced sulfur compounds such as $H_2S$, $S^0$, and $S_2O_3^{2-}$ are abundant in hydrothermal systems with hydrothermal fluids at Guaymas Basin containing aqueous $H_2S$ concentrations of up to ~6 mmol/kg (endmember measurement), while that of background seawater is negligible[47,52]. Previously reported estimates of energy budgets for sulfur-oxidizing bacteria in the Guaymas Basin hydrothermal system suggest that up to ~3900 J/kg is available for microbial metabolism, of which up to 83% may derive from sulfur oxidation[47]. Sulfur phage *dsrA* expression levels (arising from virocells) were elevated in hydrothermal systems in comparison to the background deep-sea, hinting at significant contributions of virocells mediating phage-driven sulfur oxidation to the overall energy budget. Assuming that in Guaymas Basin, the phage:total *dsrA* gene coverage ratio is 10% (the average level in Lau Basin hydrothermal environments), it may be estimated that ~320 J/kg of energy for microbial metabolism from hydrothermal vent fluids may in fact be transformed by sulfur AMGs. Although the majority of host manipulation and lysis by phages likely occurs in the absence of AMGs, we show that phages encoding sulfur AMGs can be a direct component of the sulfur biogeochemical cycle with the ability to manipulate microbial metabolism associated with multiple reduced sulfur compounds. This direct manipulation may impact sulfur budgets at ecosystem scales. It is therefore essential that future assessments of biogeochemical cycling incorporate the role of phages and their impacts on sulfur pools. Limited by the resolution of omics-based approaches in this study, finer scale phage-host interactions and activities could not be achieved, which justifies the necessity to reinforce fine-scale phage AMG activity research within host cells in the future.

Across diverse environments on the Earth, the reduced sulfur pool includes sources of deep ocean or subsurface deposited iron sulfides, and reduced sulfur species from dissimilatory sulfate reduction and organic sulfur mineralization (Fig. 8a). Sulfur phage AMG-assisted metabolism contributes to the redistribution of sulfur-generated energy and can alter its budgets, which have so far only been attributed to microbial processes (Fig. 8a). Within virocells, phage-mediated sulfur oxidation will take advantage of gene components of sulfur-metabolizing pathways, express transcripts, and produce enzymes to redirect energy for the use of phage replication (Fig. 8a). Globally distributed sulfur phages are widely distributed across various environments and impose significant impacts on the sulfur pools, as well as nutrient and energy cycling (Fig. 8a). At the same time, phage AMG mediated sulfur oxidation can short-circuit the microbial sulfur loop from

reduced sulfur pools to dissolved and particulate organic matter (DOM/POM) (Fig. 8b). Without viral infection, energy generated by reduced sulfur pools would typically be used for primary production to fuel microbial cell growth, and then transferred higher up the food chain to grazers. Through cell excretion effects, cell death and nutrient release, DOM/POM produced from sulfur-based primary production would be released to the environment. However, during infection by sulfur phages, energy generated in virocells by reduced sulfur pools could be used towards phage reproduction and propagation. After virion production and packaging, lytic phages would lyse the host cell, and release DOMs into the environment. This DSM AMG mediated approach thereby short-circuits the microbial sulfur loop. Additionally, POM generated by reduced sulfur-oxidizing processes could also be sequestered into the carbon pool deposited in the deep subsurface. It is not clear how and to what extent phage would change carbon cycling landscape between sequestrated carbon and bioavailable carbon, while it is certain that the change caused by phage AMG metabolism should be explicitly addressed in the future in the context of global biogeochemical cycle and climate change.

In conclusion, we have described the distribution, diversity and ecology of phage auxiliary metabolism associated with sulfur and demonstrated the abundance and activity of sulfur phages in the environment. Yet, many questions remain unanswered. Future research will involve unraveling mechanisms of sulfur phage and host interaction, remodeling of sulfur metabolism at the scale of individual virocells, microbial communities and ecosystems, and constraining sulfur budgets impacted by sulfur phages.

## Methods

**mVC acquisition and validation**. The Integrated Microbial Genomes and Virome (IMG/VR) database[53,54] (v2.1, October 2018) was queried for dissimilatory sulfur metabolism genes using trusted cutoffs of custom built HMM profiles[29]. A total of 192 unique mVCs greater than 5 kb in length were identified that encoded *dsr* or *sox* gene(s). For consistency between these mVCs, open reading frames were predicted using Prodigal (-p meta, v2.6.3)[55]. Each of the 192 mVCs was validated as phage using VIBRANT[56] (v1.2.1, virome mode), VirSorter[57] (v1.0.3, virome decontamination mode, virome database) and manual validation of viral hallmark annotations (Supplementary Data 5). To identify lysogenic mVCs, annotations were queried for the key terms "integrase", "recombination", "repressor" and "prophage". Annotations of validated mVCs are provided in Supplementary Data 2. Five mVCs not identified by either program were manually verified as phage according to VIBRANT annotations (i.e., KEGG, Pfam and VOG databases) by searching for viral hallmark genes, greater ratio of VOG to KEGG annotations and a high proportion of unannotated proteins. Note, not all 192 mVCs were predicted as phage by VIBRANT, but all mVCs were given full annotation profiles. One scaffold was determined to be non-viral and remove based on the presence of many bacterial-like annotations and few viral-like annotations. Validation (including software-guided and manually inspected procedures) produced a total of 191 mVCs encoding 227 DSM AMGs. It is of note that the DSM AMGs carried by three mVCs (Ga0121608_100029, Draft_10000217 and Ga0070741_10000875) could not be definitely ruled out as encoded within microbial contamination. This was determined based on the high density of non-phage annotations surrounding the AMGs in conjunction with the presence of an integrase annotation, suggesting the possibility of phage integration near the AMG.

**Taxonomy of mVCs**. Taxonomic assignment of mVCs was conducted using a custom reference database and script. To construct the reference database, NCBI GenBank[58] and RefSeq[59] (release July 2019) were queried for "prokaryotic virus". A total of 15,238 sequences greater than 3 kb were acquired. Sequences were dereplicated using Mash[60] (v2.0) and Nucmer[61] (v3.1) at 95% sequence identity and 90% coverage. Dereplication resulted in 7575 sequences. Open reading frames were predicted using Prodigal (-p meta, v2.6.3) for a total of 458,172 proteins. Taxonomy of each protein was labeled according to NCBI taxonomic assignment of the respective sequence. DIAMOND[62] (v0.9.14.115) was used to construct a protein database. Taxonomy is assigned by DIAMOND BLASTp[63] matches of proteins from an unknown phage sequence to the constructed database at the classifications of Order, Family and Sub-family. Assignment consists of reference protein taxonomy matching to each classification at the individual and all protein levels to hierarchically select the most likely taxonomic match rather than the most common (i.e., not recruitment of most common match). Taxonomic assignments are available for 25 Families and 29 Sub-families for both bacterial and archaeal

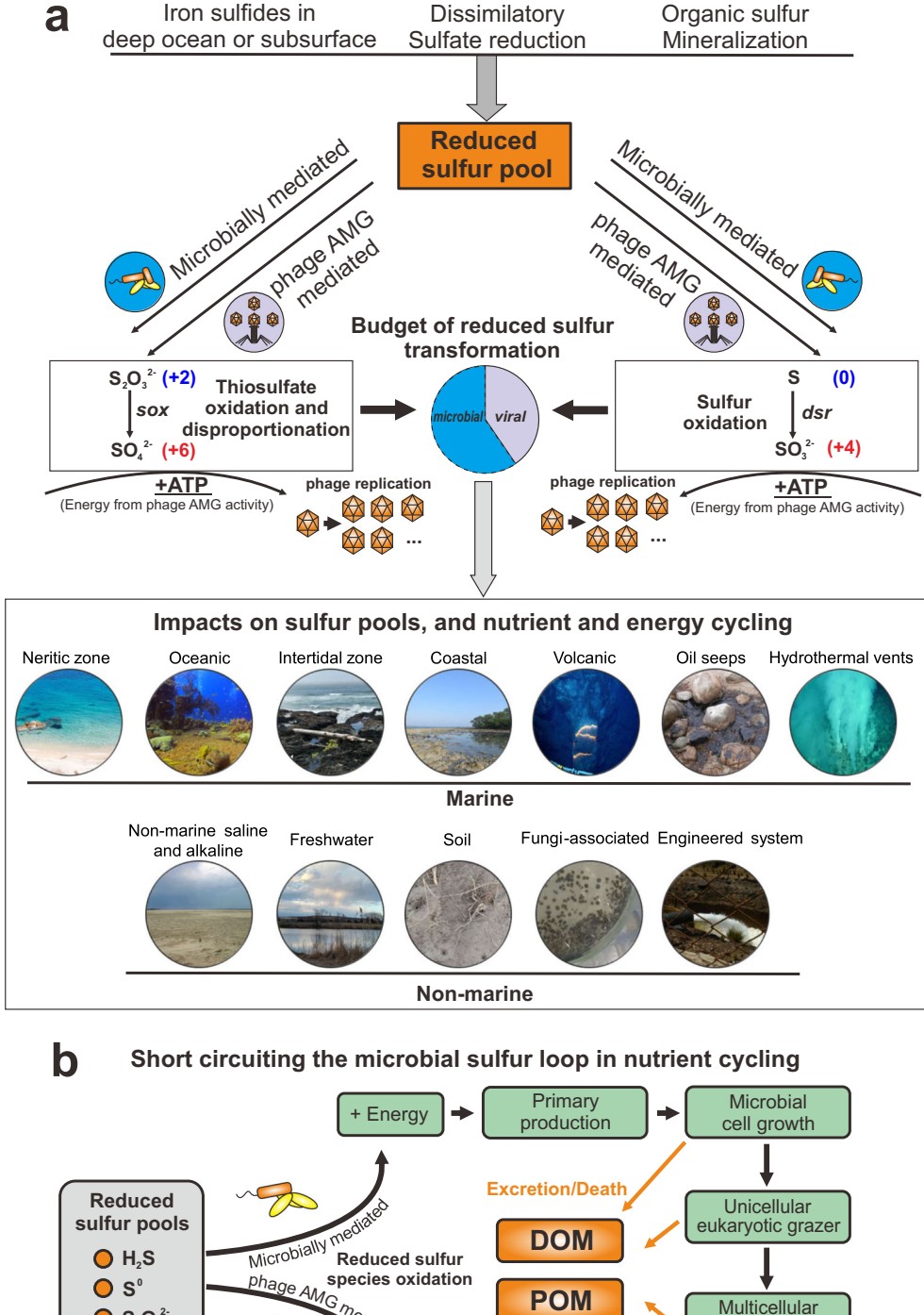

**Fig. 8 Conceptual figure indicating the ecology and function of AMGs in sulfur metabolisms. a** DSM AMG effect on the budget of reduced sulfur transformation. **b** Diagram of virus-mediated metabolism short circuiting the microbial sulfur loop in nutrient cycling. DOM: dissolved organic matter, POM: particulate organic matter.

viruses. The database, script and associated files used to assign taxonomy are provided. To construct the protein network diagram vConTACT2[64] (v0.9.5, default parameters) was used to cluster mVCs with reference viruses from NCBI from the families *Ackermannviridae, Herelleviridae, Inoviridae, Microviridae, Myoviridae, Podoviridae,* and *Siphoviridae* as well as several archaea-infecting families. The network was visualized using Cytoscape[65] (v3.7.2) and colored according to family affiliation.

**Host prediction and CRISPR spacer analysis.** A total of 25 representative metagenomes containing putative host sequences were downloaded from IMG (3300001676, 3300001678, 3300001679, 3300001680, 3300001681, 3300001683, 3300007516, 3300007722, 3300009154, 3300009182, 3300010157, 3300010296, 3300010297, 3300010299, 3300010300, 3300010318, 3300010354, 3300010370, 3300020258, 3300020264, 3300020266, 3300020314, 3300020325, 3300020365, 3300020454). Metagenome sequences were limited to a length of 10 kb (149,986 total sequences). CRISPR Recognition Tool (CRT, v1.2, default settings)[66] was used to identify 7,178 CRISPR spacers from the 149,986 putative host sequences. Blastn (v2.2.31) was used to search the 191 mVC genome sequences for alignment to the spacers. A spacer hit was considered positive with 100% coverage to the spacer and 0–2 mismatches. To validate that the method worked properly, the 7178 spacers were used to query the entire IMG/VR database (v2.1, October 2018).

**World map distribution of mVCs.** IMG/VR Taxon Object ID numbers respective of each mVCs were used to identify global coordinates of studies according to IMG documentation. Coordinates were mapped using Matplotlib (v3.0.0) Basemap[67] (v1.2.0). Human studies were excluded from coordinate maps.

**Sequence alignments and conserved residues.** Protein alignments were performed using MAFFT[68,69] (v7.388, default parameters). Visualization of alignments was done using Geneious Prime 2019.0.3. N- and C-terminal ends of protein alignments were manually removed, and aligned columns with 90% (SoxD and SoxYZ) or 98% (DsrA and DsrC/TusE) gaps were stripped (masked) for clarity. Amino acid residues were highlighted by pairwise identity of 90% (SoxC and SoxYZ) or 95% (DsrA, DsrC/TusE and SoxD). An identity graph, generated by Geneious, was fitted to the alignment to visualize pairwise identity of 100% (green), 99–30% (yellow), and 29–0% (red). Conservation of domains and amino acid residues was assessed according to annotations by The Protein Data Bank.

To calculate d$N$/d$S$ ratios between mVC AMG pairs, dRep[70] (v2.6.2) was used to compare AMG sequences of *dsrA* ($n = 39$), *dsrC* ($n = 141$), and *soxYZ* ($n = 44$) separately (dRep compare–SkipMash–S_algorithm goANI). A custom auxiliary script (dnds_from_drep.py[71]) was used to calculate d$N$/d$S$ ratios from the dRep output between various AMG pairs. Resulting d$N$/d$S$ values were plotted using Seaborn[72] (v0.8.1) and Matplotlib. Phage AMG pairs and respective d$N$/d$S$ values can be found in Supplementary Data 6.

**mVC protein grouping.** All protein sequences of 94 mVCs, excluding those with non-validated DsrC (i.e., potentially TusE-like) AMGs according to the conserved CxxxxxxxxxxC motif, were grouped using mmseqs2[73] (–min-seq-id 0.3 -c 0.6 -s 7.5 -e 0.001). For the AMG neighbor protein grouping, a total of 70 mVCs were used that encoded at least five proteins before and five proteins after the AMG. Groups containing at least two different representative mVCs were retained (887 groups total). A presence/absence heatmap was made using the R package "ComplexHeatmap"[74] and hierarchically grouped according to the ward.D method. Metadata for AMG, taxonomy and source environment were laid over the grouped columns. Two mVCs, Ga0066448_1000315 and JGI24724J26744_10000298, were not represented by any of the 887 retained clusters. mVC alignments were done using EasyFig[75] (v2.2.2).

To validate that our grouping method accurately depicts whole genome diversity of the mVCs, *Caudovirales* phages from the NCBI RefSeq database were used as a comparison. All *Caudovirales* phages were downloaded and dereplicated by 97% identity and 90% coverage using Mash and Nucmer. The dereplicated set consisted of 4413 RefSeq phages. A total of 94 RefSeq phages were randomly selected. Proteins were predicted using Prodigal and all proteins were grouped as before (mmseqs2–min-seq-id 0.3 -c 0.6 -s 7.5 -e 0.001, minimum group size of 2 members). Random phage selection and protein grouping was performed 100 independent times (iterations). Over the 100 iterations the 94 randomly selected phages encoded 6821 to 10,123 proteins (average 8357) and generated 727 to 1289 protein groups (average 989). The number of clusters per protein ranged from 0.088 to 0.149 (average 0.119). These statistics were similar to those seen from the mVCs, which generated 794 clusters from 6015 encoded proteins. The number of clusters per protein was 0.132. Therefore, despite encoding fewer proteins on average compared to RefSeq *Caudovirales*, the mVCs generated a comparable number of clusters per protein (Supplementary Data 7).

**mVC genome structure and organization.** mVCs representative of each AMG family were selected. Annotations were performed using VIBRANT and the best scoring annotation was used. Genomes were visualized using Geneious Prime and manually colored according to function.

**AMG protein phylogenetic tree reconstruction.** DSM protein sequences from reference prokaryotes were downloaded from NCBI nr database (accessed May 2019). The results were manually filtered for accurate annotations. The curated results were clustered by 70% sequence similarity using CD-HIT[76] (v4.7). These representative sequences from individual clusters were aligned with the corresponding mVC AMG protein sequences using MAFFT (default settings). Alignments were subjected to phylogenetic tree reconstruction using IQ-TREE[77] (v1.6.9) with the following settings: -m MFP -bb 100 -s -redo -mset WAG,LG,JTT,Dayhoff -mrate E,I,G,I + G -mfreq FU -wbtl ("LG + G4" was chosen as the best-fit tree reconstruction model). The environmental origin information of each mVC AMG was used to generate the stripe ring within the phylogenetic tree in the operation frame of iTOL[78] online server.

**Metagenomic mapping and gene coverage ratio calculation.** The metagenomic reads were first dereplicated by a custom Perl script and trimmed by Sickle[79] (v1.33, default settings). The QC-passed metagenomic reads were used to map against the collection of genes of investigated metagenomic assemblies by Bowtie2[80] (v2.3.4.1). The gene coverage for each gene was calculated by "jgi_summarize_bam_contig_depths" command within metaWRAP[81] (v1.0.2). The phage:total gene coverage ratio was calculated by adding up all the phage and bacterial gene coverage values and using it to divide the summed phage gene coverage values.

We identified the phage-host gene pairs in the phylogenetic tree containing AMG and their bacterial counterpart gene encoding proteins. We assigned the phage-host gene pairs according to the following two criteria: (1) The phage and host gene encoding proteins are phylogenetically close in the tree; the branches containing them should be neighboring branches. (2) They should be from the same metagenomic dataset, which means that AMGs and bacterial host genes are from the same environment sample. The identified phage-host gene pairs were labeled accordingly in the phylogenetic tree.

For the gene coverage ratio calculation of phage genes and bacterial genes within a phage-host pair, we first calculated the phage:total gene coverage ratio and bacterial:total gene coverage ratio using the same method as described above; and then, in order to avoid the influence of numbers of phage or bacterial genes, we normalized the above two ratio values by the number of phage and bacterial genes, respectively. Finally, the normalized phage:host gene coverage ratio of this phage-host pair was calculated by comparing these two ratio values, accordingly.

Additionally, reads mapping performance was re-checked by comparing original mapping results (using Bowtie 2 "-very-sensitive" option) to the mapping results that only include reads with one mismatch (Supplementary Fig. 8). Checking results have justified the reliability of our original mapping performance and our gene coverage ratio calculation.

**Metatranscriptomic mapping.** The metatranscriptomic reads were first dereplicated by a custom Perl script and trimmed by Sickle (default settings), and then subjected to rRNA-filtering using SortMeRNA[82] (v2.0) with the 8 default rRNA databases (including prokaryotic 16S rRNA, 23S rRNA; eukaryotic 18S rRNA, 28S rRNA; and Rfam 5S rRNA and 5.8S rRNA). QC-passed metagenomic reads were mapped against the collection of AMGs using Bowtie2 (–very-sensitive). The gene expression level in Reads Per Kilobase per Million mapped reads (RPKM) was calculated by normalizing the sequence depth (per million reads) and the length of the gene (in kilobases).

**Reporting summary.** Further information on research design is available in the Nature Research Reporting Summary linked to this article.

## Data availability

All IMG/VR sequences are available at https://img.jgi.doe.gov/cgi-bin/vr/main.cgi and https://genome.jgi.doe.gov/portal/pages/dynamicOrganismDownload.jsf?organism=IMG_VR. Sequences from identified mVCs are available publicly and described in Supplementary Data 1 and 2. Any other relevant data are available from the authors upon request.

## Code availability

All sequences and custom analysis scripts used in this study are also available at https://github.com/AnantharamanLab/Kieft_and_Zhou_et_al._2020.

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

## Acknowledgements

We thank the University of Wisconsin—Office of the Vice Chancellor for Research and Graduate Education, University of Wisconsin—Department of Bacteriology, and University of Wisconsin—College of Agriculture and Life Sciences for their support. K.K. is supported by a Wisconsin Distinguished Graduate Fellowship Award from the University of Wisconsin-Madison. This work was partly supported by the National Science Foundation grants OCE-2049478 to K.A. and OCE-0961947 to M.B.S. The work conducted by the U.S. Department of Energy Joint Genome Institute is supported by the Office of Science of the U.S. Department of Energy under contract no. DE-AC02-05CH11231.

## Author contributions

K.K., Z.Z., S.R., and K.A. designed the study. K.K. and S.R. identified the genomes. K.K., Z.Z., and K.A. conducted the analyses. K.K., Z.Z., and K.A. drafted the manuscript. All authors (K.K., Z.Z., R.E.A., A.B., B.J.C., S.J.H., M.H., M.B.S., D.A.W., S.R., and K.A.) reviewed the results, revised, and approved the manuscript.

## Competing interests

The authors declare no competing interests.
