## [Peer Review File · Nature Communications]

Reviewers' comments:

Reviewer #1 (Remarks to the Author):

Kieft et al present an analysis of sulfur auxiliary metabolic genes (AMGs) encoded by bacteriophage present in the IMG/VR database. The authors annotate genes belonging to the *dsrA*, *dsrC/tusE* and *soxCDYZ* families, and provide some gene phylogenies and genomic comparisons of the phage sequences that encode these genes. They also analyze some metatranscriptomes to show that these genes are expressed in some cases.

The authors do not present new metagenomes, and this study is an analysis of sequences available in the IMG/VR database. It is good that researchers use IMG/VR for these purposes, but the novelty of this study is limited since this database has been available for years. No novel approaches for analyzing the existing data are presented. Moreover, most of the genes described in this study have already been reported in bacteriophage, so most researchers will not be surprised by this finding. The main value of this study is that it provides a more complete survey of sulfur metabolism genes in phage, even though the results are not substantially different than those reported in other studies (Anatharaman, 2014, Science PMID: 24789974, Roux, Nature, 2016 PMID: 27654921). The finding of SoxC and SoxD in phage is new, and the combined description of sulfur related AMGs will be useful to some researchers. The analysis of active site residues are also valuable and will also be interesting to some specialists in this field, but even these findings are contradictory to the main claims of the study, and motif analysis of some sulfur AMGs has also been reported previously (Roux, Nature, 2016). These advances are incremental and narrow in scope compared to what has been published before. The findings of this study cannot be considered a significant advance given so much has already been reported.

A major conclusion of the study is that these AMGs show that bacteriophage play an important role in sulfur cycling. This conclusion is speculative since only omic evidence is presented here and no actual biogeochemical measurements have been made, and the hosts of the phage are unknown. Based on the presence of these genes alone it is unclear if the proteins are highly active, or if sulfur cycling is even particularly prevalent in some of the environments studied. Although broad surveys sound exciting because of the implied global importance, they also lack ecosystem-specific context which is critical for evaluating the results.

One of the most interesting findings of this study is that many AMGs did not have the conserved functional motifs that would be expected for active sulfur metabolism genes. This suggests that these genes have alternative functions that are unknown. Given this is one of the more novel aspects of this study it should be emphasized in the discussion section. Right now most of the discussion assumes the enzymes have the typical activity, a claim that is contradicted by the data.

Some of the most exciting work in virocell biology has recently been done in model systems where host-phage interactions can be assessed in detail (Howard-Verona, 2016 ISMEJ, PMID: 27187794 - Howard-Varona, 2020, ISMEJ PMID: 31896786). I encourage the authors to more clearly discuss these complications and limitations so that readers will have a more balanced view of how phages manipulate host metabolism (one of these papers is discussed in the intro but only as a general reference for virocells).

The 190 metagenome-assembled viruses appear to actually be viral contigs, not complete genomes, and some are quite short (5 kbp) and the average is only 31 kbp. When examining such short contigs it is difficult to understand the biological context of the enzymes or the clades of phage that encode them. Some genome plots are presented, but they are difficult to interpret given their short length. It is also possible some have been misclassified as viral, since accurate classification is difficult for short sequences. In depth characterization is only really possible for the largest contigs found here, but even they are likely not complete. Some important work on bacteriophage biology has recently been

done by the Banfield group, which has meticulously examined complete genomes of phage (some reaching > 700 kbp) (Al-Shayeb, 2020 Nature, PMID: 32051592). When AMGs are found in these complete phages a much more in-depth analysis can be done, since analysis of the complete phage genome allows for much more detailed understanding of the evolutionary history, clades, predicted host, and ecological niches of the virus.

The 190 viral contigs of sulfur phages comprise a small proportion of IMGVR, which contains hundreds of thousands of contigs. So less than 0.01% of the contigs had sulfur metabolism genes- this statistic should be calculated exactly and provided to put these findings in context. It seems that these AMGs are rare and found in specific environments, so the claims of the importance of sulfur phage for global elemental cycles are overstated and should be qualified.

Although AMGs may be involved in biogeochemical cycles, viruses do not need these enzymes to have important environmental impacts. Arguably the major influence of bacteriophage on elemental cycles is their killing of host cells. Other phage that do not encode AMGs could therefore have a much bigger impact on sulfur cycling than those described here. The AMGs are potentially only active for short periods of time during a brief window of the infection, so gene or transcript abundances may give a misleading view of their importance, especially for isolated metagenomes/transcriptomes. I encourage the authors to include these caveats in their study. Right now it reads a bit like AMGs are the only way phages can shape elemental cycles.

Little information on the hosts infected by these sulfur phages is given, which limits the ability to infer any potential biogeochemical impact. Understanding the host is critical to evaluating the environmental role of the virus, but that is not addressed in this study. Recent work examining methane metabolism AMGs in freshwater phage was able to reconstruct complete genomes of novel phage and provided in-depth links to predicted hosts, for example, so a more complete picture of how phage influence biogeochemical cycles was provided (Chen, 2020, Nature Microbiology PMID: 32839536). To improve this study the authors could perform CRISPR or co-occurrence analyses of the phage they found to identify putative hosts. Given the diverse hosts involved in sulfur cycling it is likely that the AMGs have different roles depending on the host, and this possibility should be discussed.

Figures 2 and 8 are very general and summarize broad aspects of phage infection and elemental cycling that are already known. In general the manuscript could be shortened to focus on the novel aspects of this study (detailed phylogenies on all marker genes identified) and less on that which has already been reported by other studies.

Groups such as podoviridae, myoviridae, and siphoviridae are defined primarily by morphology, so family level classifications cannot be accurately made based on genomic data alone. The classifications of many phage in NCBI are incorrect, so these errors will be propagated here. Many of these families appear mixed together in Figure 4, and it is unclear if any of the data is supported by actual microscopy.

The network figure is general and difficult to interpret, especially given the problems with viral taxonomy and incomplete genomes. It is difficult to interpret the finding that sulfur phage cluster with known phage of pelagibacter and cyanobacteria. It seems like this is artefact. One can only conclude that the sulfur phage sometimes cluster with known phage. A general summary figure showing how many contigs were similar to known phage would be more useful.

The section titled "Estimates of sulfur phage contributions to sulfur oxidation" should be renamed since no actual measurements of sulfur oxidation have been made. This section appears to be based on gene abundance ratios, which are very different.

Reviewer #2 (Remarks to the Author):

In this manuscript, Kieft, Zhou, et al. study the ecology, metabolism, and distribution of phages that infect sulfur species and carry AMGs involved in sulfur metabolism. The authors take advantage of a dataset consisting of 191 phage scaffolds reconstructed from metagenomes that contain phage AMGs related to sulfur metabolism. The authors identify two previously unknown sulfur-related AMGs and present multiple (and, to the best of my knowledge, novel) analyses that put sulfur AMGs in environmental, ecological, and host-phage interaction contexts. These include identifying the potential role of sulfur AMGs in virocell metabolism, evaluating the abundance of sulfur phages and sulfur AMGs across different environments, determining sulfur phage taxonomy and host range, and more. Overall the manuscript is well written. The analyses presented are insightful and of high quality, and the authors make good use of public data. The paper could be of interest to researchers interested in microbiology, sulfur metabolism, and biogeochemical cycles (sulfur in particular, of course).

- "As many as 30-40% of all bacteria are assumed to be in a virocell state, undergoing phage-directed metabolism" (lines 69-70): in Curtis Suttle's paper, it is stated that "viral lysis in surface waters removes 20-40% of the standing stock of prokaryotes each day". Lysing is different from being in a virocell state. If 20-40% of the cells are lysed every day, I would assume that the share of virocells from the total prokaryotic cells is significantly lower, depending on the estimated average phage latent period. The share of virocells is important because it is used later (456-460).

- "We identified 190 viral metagenome-assembled genomes (vMAGs) and one viral single-amplified genome" (lines 123-124): The use of the term "vMAGs" is a bit of a stretch in this case, "viral scaffolds" may be more appropriate. The majority of scaffolds (108/191) are shorter than 20Kbp, which is roughly half the size of a podovirus genome. Considering that most of the scaffolds are predicted to represent myovirus genomes, which are much longer, the majority of scaffolds are probably far from representing genomes.

- "Most vMAGs clustered with reference phages that infect Pelagibacter, Cyanobacteria and Enterobacteria ... " (283-287): it makes sense to me to include the information available in Supp. Fig. S8-S11 when discussing the potential hosts of the sulfur phages. I assume the information in the phylogenetic trees is more reliable for this purpose.

- "Only a few protein groups were globally shared amongst the vMAGs, including common phage proteins" (294-295): Could this be the result of the partial genomes available for the phages? 132 vMAGs were identified as myoviruses, which typically have long genomes, at the range of >100Kbp. However, there are only 9 vMAGs longer than 100Kbp. Since the dataset consists only of scaffolds with sulfur AMGs it is likely that many other phage genes are missing.

- "Conservatively assuming that 40% of all sulfur-oxidizing SUP05 Gammaproteobacteria are infected by sulfur phages (in line with observations of phage infections in the pelagic oceans) ..." (lines 457-460): see above comment regarding the 40% figure. Also: sulfur-oxidizing SUP05 Gammaproteobacteria may be infected by phages that do not contain sulfur AMGs and therefore do not affect the host's sulfur metabolism. Therefore, the energy estimate seems too high (or the assumptions need to be explained in more detail).

Minor comments:

- Fig. 1 (also possible in the text): how does the organization of the sulfur genes on the phage genomes compare to their organization on the genomes of their potential hosts?
- Fig. 6b, 7b: TARA_B100000034 appears twice, is this intentional?
- Sentence in lines 453-456 may be incomplete?

The review was written by Itai Sharon.

Reviewer #3 (Remarks to the Author):

Kieft and Zhou et al. have followed up on earlier work identifying sulfur-cycling genes in phages to demonstrate how widespread this phenomenon is. There appears to be an higher representation of these genes in phages in certain environments than one would expect by pure chance. These metabolic genes are highly expressed, and a phage introducing extra gene copies would have an impact on the final amount of protein produced for rate-limiting steps and provide a fitness advantage for the organism. It should be noted that this and earlier work in this field is still somewhat speculative - there doesn't seem to have been any experiments confirming this supposition in laboratory cultures. However, the quantitative evidence from metagenomes (in particular figures 6 and 7) is hard to argue with and it seems that the authors are really onto something here.

The only major criticism that I have with this work is that the authors examine a lot of different environments without taking into account whether or not there is any sulfur cycling (sulfate reduction/sulfide oxidation) there. One would expect very little sulfur-cycling in fully oxygenated aquatic environments, for example. Which of the Tara Ocean samples were from OMZs, for example? Wouldn't one expect anoxic environments with more selection for sulfate reduction to have higher phage:host ratios? So a more nuanced discussion of the geochemical conditions in the environments where this data (especially quantitative data in figures 6 and 7) comes from could be helpful, with any quantitative patterns relating to the predicted degree of sulfur cycling highlighted.

line 69-70: "of all bacteria" is a very broad statement - this should be qualified to be limited to the environments investigated in the citations 10 and 11.

lines 73-77: More of an explanation for how these AMGs work would be helpful here. It is critical that these genes are highly expressed central metabolic genes encoding rate-limiting steps - otherwise adding an additional copy on the phage genome would not provide any fitness advantage. The other lingering question is why hasn't evolution simply boosted the copy number on the host's genome? This suggests that phage infection only boosts fitness sometimes. A more in-depth discussion of these nuances would help a reader like me coming from outside the phage AMG field trying to understand how it all fits together.

line 96: "and" should not be italicized

line 128: this statement needs a citation.

Figure 2: The way these figures are drawn it seems that phage infection takes the bacteria from having no DsrAC or SoxYZ/CD to having it, whereas as I understand it the model is that the phage genes augment the genes with the same function that the hosts already have? It would be clearer if the figures were modified to show that this is an augmentation rather than an initial introduction of the genes.

Figure 2: The chemical reactions in the boxes are confusing - the "complete reactions" are half reactions and the "reaction catalyzed by" need other enzymes than those mentioned. Chemical stoichiometry is not relevant to the story so they could easily be removed without detriment to the manuscript.

Figure 6: Why aren't all genes being shown for all environments? Are the ones not shown just 0% (no dsrA or soxYZ on any phages at all)?

Line 322: "Estimates of sulfur phage contributions to sulfur oxidation" - this title seems to over-

promise a bit. Wouldn't the fraction of infected cells be necessary knowledge to truly estimate the contribution to sulfur oxidation? That is not data that is available in this study, isn't it?

lines 382-392: There are several places in this paragraph where genes are referred to, e.g. line 385 "phage to host dsrA gene ratios", when it seems the authors mean to refer to gene transcripts?

line 387: So infinity means that there was zero host gene coverage? Maybe just say that?

Figure 7AB: Why are these data shown as ratios, while they are shown as percentages in figure 6? Shouldn't this be consistent throughout the paper?

line 411: Why is it likely that DsrA has a greater affect than DsrB? This statement needs some degree of justification.

line 417: "is provide" should be "is provided"

line 458: More details for this calculation is needed - what numbers are being used for cell numbers and rates here?

line 477 and figure 8b: It's important to keep in mind that a lot of biomass created in these environments will end up buried in the deep subsurface and turned over very slowly, rather than entering the food chain, thus contributing to global controls on oxygen and CO₂. Could these phages be helping to regulate the global climate? That might be an important point to include in the discussion and figure 8.

line 494-495: Please specify how the IMG/VR database was queried for dsr and sox. Was this HMMer? BLAST? Text-based search? KEGG? COG? What was it?

line 497: Does this mean that all 192 passed screening with these tools, or was there a larger number and some were discarded?

line 504: "not all vMAGs were predicted as phage" - does this refer to the five vMAGs talked about on line 501? Or others?

line 540: "gaps were stripped by 90% (SoxD and SoxYZ) or 98% (DsrA and DsrC/TusE)" - is there another way of expressing this? It seems unclear what was done with the gaps.

Ian Marshall

Karthik Anantharaman
Department of Bacteriology
University of Wisconsin-Madison
1550 Linden Drive
Madison, WI 53706-1567

Ph: 608-265-4537
Fax: 608-262-9865
Email: karthik@bact.wisc.edu

May 6, 2021

Reviewers' comments:

Reviewer #1 (Remarks to the Author):

Kieft et al present an analysis of sulfur auxiliary metabolic genes (AMGs) encoded by bacteriophage present in the IMG/VR database. The authors annotate genes belonging to the *dsrA*, *dsrC/tusE* and *soxCDYZ* families, and provide some gene phylogenies and genomic comparisons of the phage sequences that encode these genes. They also analyze some metatranscriptomes to show that these genes are expressed in some cases.

Reply: We thank the reviewer for their detailed comments.

- The authors do not present new metagenomes, and this study is an analysis of sequences available in the IMG/VR database. It is good that researchers use IMG/VR for these purposes, but the novelty of this study is limited since this database has been available for years.

- **Reply:** We disagree with the premise that old datasets do not contain novel information. Within his/her review, the reviewer has cited Al-Shayeb et al., 2020 *Nature* (PMID: 32051592) as a model paper for describing a 700kb phage. This manuscript utilized both the *Tara Oceans* and *Global Ocean Virome* databases which are also years old (2011 and 2016, respectively). These databases (IMG/VR, *Tara Oceans*, GOV) are constructed as repositories of data for the discovery of new and useful information. The recent manuscripts "A new view of the tree of life" (Hug et. al., 2016, PMID: 27572647, *Nature Microbiology*) as well as "A genomic catalog of Earth's microbiomes" (Nayfach et. al., 2020, PMID: 33169036, *Nature Biotechnology*) likewise utilized publicly available data (e.g., IMG/M) and are widely accepted in the field. To use genomes from publicly available databases for addressing scientific questions should not be considered as lacking novelty. Novelty roots in how to interpret biological and ecological outcomes from datasets and whether the results are scientifically remarkable and insightful. Since the rapid increase of data sizes and development of new methods, many scientists have leveraged growing-size of datasets to conduct novel and insightful research.

- No novel approaches for analyzing the existing data are presented.

- **Reply:** Multiple studies discussed by the reviewer (Chen et al., 2020, Al-Shayeb et al., 2020) to refute the impact of the data presented here contain no novel approaches. Beyond CRISPR spacer analysis to predict hosts, which we have now implemented, the two studies follow similar methodology for studying phages encoding AMGs (transcriptomics, AMG phylogeny, phage phylogeny, environmental nutrient context, conceptualization of AMG impact, etc.). The reasoning for this is straightforward: the methodology used in these two

articles, in addition to the methods displayed in the data presented here, reflect the most state-of-the-art methods for metagenomic phage analysis. The reviewer has expected that this manuscript presents both novel information (i.e., expanding known DSM AMG on phages) as well as generates new methodology, which was not the scope of the study.

- Moreover, most of the genes described in this study have already been reported in bacteriophage, so most researchers will not be surprised by this finding. The main value of this study is that it provides a more complete survey of sulfur metabolism genes in phage, even though the results are not substantially different than those reported in other studies (Anantharaman, 2014, Science PMID: 24789974, Roux, Nature, 2016 PMID: 27654921). The finding of SoxC and SoxD in phage is new, and the combined description of sulfur related AMGs will be useful to some researchers. The analysis of active site residues are also valuable and will also be interesting to some specialists in this field, but even these findings are contradictory to the main claims of the study, and motif analysis of some sulfur AMGs has also been reported previously (Roux, Nature, 2016). These advances are incremental and narrow in scope compared to what has been published before. The findings of this study cannot be considered a significant advance given so much has already been reported.

- **Reply:** We believe the outcome of this research is not only for specialists but conversely has broad impact for the whole academic community. Beyond sulfur phage AMGs firstly discovered in the mentioned publications, we discovered that AMGs encoding sulfur carriers (*soxYZ* and *dsrC*) rather than catalytic/enzymatic subunits appear to be favored by phages, contrary to previous thoughts. Our research is not limited to reporting new sulfur phage AMGs, but also is expanded to investigate their evolutionary mechanisms and ecological impacts. In this study, we used evolutionary analyses to demonstrate retention of AMGs during niche-differentiation of diverse phage populations which provides evidence that auxiliary metabolism imparts measurable fitness benefits to phages with ramifications for ecosystem biogeochemistry. Moreover, we utilized multiple-omics analyses to reveal significant contributions by phages to sulfur and thiosulfate oxidation in hydrothermal environments. These contributions and implications go beyond what has been previously reported.

Based on analyzing these sulfur phage AMGs and relevant metagenomic and metatranscriptomics datasets, the evolutionary and ecological findings described above have also provided novel insights into the phage-associated dissimilatory sulfur metabolisms, which highlights the necessity of taking viruses into the configuration of metabolism and biogeochemistry in an integrated manner.

Our study follows in the footsteps of reports that described AMGs associated with important biogeochemical processes including photosynthesis, sulfur/methane/ammonia oxidation, carbon utilization, etc. (Nature 424(6950):741; PNAS. 2011 108(39); Anantharaman et al., Science. 2014 344(6185):757-60; ISME J. 2019 (3):618-631), which are inspiring fundamentally new avenues of research and a wealth of publications. We believe the findings of this study is also equally significant and provides scientific advances in this field, and will be of interest to a wide range of audiences of microbiology, ecology, and biogeochemistry background.

- A major conclusion of the study is that these AMGs show that bacteriophage play an important role in sulfur cycling. This conclusion is speculative since only omic evidence is presented here and no actual biogeochemical measurements have been made, and the hosts of the phage are

unknown. Based on the presence of these genes alone it is unclear if the proteins are highly active, or if sulfur cycling is even particularly prevalent in some of the environments studied. Although broad surveys sound exciting because of the implied global importance, they also lack ecosystem-specific context which is critical for evaluating the results.

- **Reply:** We admit that we do not have any model systems to study these AMGs in the lab or make any measurements of biogeochemistry, this is due to factors beyond our control or scope of this study such as the difficulty in cultivation of the hosts of these phages. In fact, no cultured system exists with a phage that encodes a DSM AMG. Moreover, many model papers for phage AMGs (such as methane AMGs in Chen, 2020, Nature Microbiology PMID: 32839536) do not include biogeochemical measurements to verify phage impacts.

- One of the most interesting findings of this study is that many AMGs did not have the conserved functional motifs that would be expected for active sulfur metabolism genes. This suggests that these genes have alternative functions that are unknown. Given this is one of the more novel aspects of this study it should be emphasized in the discussion section. Right now most of the discussion assumes the enzymes have the typical activity, a claim that is contradicted by the data.

- **Reply:** We have added discussion of this topic at lines 469-477. In this we provide an example of a cyanophage that also encodes a divergent AMG as precedent for phages retaining biochemical function of AMG-encoded proteins despite altering key protein domains.

- Some of the most exciting work in virocell biology has recently been done in model systems where host-phage interactions can be assessed in detail (Howard-Verona, 2016 ISMEJ, PMID: 27187794 - Howard-Varona, 2020, ISMEJ PMID: 31896786). I encourage the authors to more clearly discuss these complications and limitations so that readers will have a more balanced view of how phages manipulate host metabolism (one of these papers is discussed in the intro but only as a general reference for virocells).

- **Reply:** Both manuscripts presented here (27187794 and 31896786) detail transcriptomic and proteomic evidence of phage:host interactions within a model system. Both manuscripts likewise describe infection in terms of known infection conditions (e.g., temperature, lysis time, latency, multi-time point transcriptomic and proteomic data, and other information yielded from model systems such as these). These two presented studies are quite different from our methodology of metagenomic surveys of phage AMGs and are thus difficult to compare. We acknowledge that model systems can provide much more detailed information regarding the mechanisms of phage manipulation of host metabolism, but these studies are beyond the scope of this research as we have no cultured representatives of sulfur microbe infecting phages that encode DSM AMGs. Thus, metagenomic surveys such as the one presented here are currently the only method for elucidating the role of sulfur AMGs in the environment. To acknowledge this in the manuscript we provided details of how model systems would be beneficial for determining the effect of sulfur AMGs during infection and how our study lacks this evidence (lines 469-477).

Excerpt from the Discussion (line 465): “Although the fitness effects of DSM AMGs have not been quantified in a model system, the geographical distribution of identified mVCs and retention of AMGs by phages despite constrained coding capacity strongly suggests a significant fitness benefit of encoding DSM AMGs. The exact fitness benefit achieved

from encoding DSM AMG remains elusive without cultured representatives of phage-host pairs.”

- The 190 metagenome-assembled viruses appear to actually be viral contigs, not complete genomes, and some are quite short (5 kbp) and the average is only 31 kbp. When examining such short contigs it is difficult to understand the biological context of the enzymes or the clades of phage that encode them. Some genome plots are presented, but they are difficult to interpret given their short length. It is also possible some have been misclassified as viral, since accurate classification is difficult for short sequences. In depth characterization is only really possible for the largest contigs found here, but even they are likely not complete. Some important work on bacteriophage biology has recently been done by the Banfield group, which has meticulously examined complete genomes of phage (some reaching > 700 kbp) (Al-Shayeb, 2020 Nature, PMID: 32051592). When AMGs are found in these complete phages a much more in-depth analysis can be done, since analysis of the complete phage genome allows for much more detailed understanding of the evolutionary history, clades, predicted host, and ecological niches of the virus.

- **Reply:** We have provided clarification within the text (line 135, 149, 315, 323) that the phages presented here represent partial genome scaffolds rather than whole genomes. However, we conducted and provided evidence for our rigorous evaluation of these scaffolds as phages rather than bacterial contamination (Methods, line 547) Although we provide caveats of the phage lengths, we are confident they are sufficient for the analyses presented here, especially since the focus was on the individual AMGs. As a test of average viral genome size, all prokaryotic (bacterial and archaeal) viruses/phages were downloaded from NCBI in July 2019. The sequences were filtered to only include complete genomes which resulted in a total of 13,541 resulting viruses/phages.

a. Results: The minimum and maximum genome sizes were 3,037bp and 551,627bp, respectively. The average and median genome sizes were 55,452bp and 42,723bp, respectively. The claim that 31kb represents only a short genome fragment is misfounded as it is ~73% of the median length of all complete prokaryotic viral genomes on NCBI.

b. The idealization that >700kb is best for viral classification is likely a misunderstanding of prokaryotic viral genomes. It is not rare to identify eukaryotic giant viruses >1Mbp, but this is yet to be discovered for prokaryotic viruses and identifying a genome of 700kb is extraordinarily rare. First, the largest prokaryotic (archaeal and bacterial) viral genome deposited on NCBI is ~550kb. Only a total of 56 (0.4%) viral genomes are greater than 300kb and only 278 (2%) are above 200kb. Conversely, 4,227 (31%) are shorter than 31kb and 1,367 (10%) are shorter than 5kb. Therefore, it's more unlikely to obtain a large (>200kb) viral genome than it is to obtain one less than 30kb, especially for non-isolate (e.g., metagenomic) viruses.

c. Viral length statistics for the IMG/VR database used in the study: 16kb average, 630kb maximum, 5kb minimum, 11kb median. Again, the average length of 31kb does not denote fragments that are too short to analyze.

- The 190 viral contigs of sulfur phages comprise a small proportion of IMGVR, which contains hundreds of thousands of contigs. So less than 0.01% of the contigs had sulfur metabolism genes- this statistic should be calculated exactly and provided to put these findings in context. It seems that these AMGs are rare and found in specific environments, so the claims of the importance of sulfur phage for global elemental cycles are overstated and should be qualified.

- **Reply:** The small percentage of ‘rare’ phage *dsrA* genes in the database does not underestimate the importance of sulfur-oxidizing bacteria or phages with DSM AMGs in

the global biogeochemical cycling process. To highlight this we analyzed the abundance of sulfur oxidizing bacteria in the IMG/M database (microbial equivalent to IMG/VR). We searched ~56 million bacterial scaffolds of which only ~1000 scaffolds encoded *dsrA* genes (0.0018%). Despite this small percentage, bacteria encoding *dsrA* genes are known to significantly contribute to global sulfur biogeochemistry and cycling.

In a recently published paper (Chen, 2020, Nature Microbiology PMID: 32839536), viruses containing methane oxidizing AMGs were found to occupy a low portion in the total community (0.0000218%). Despite this, it is hypothesized that these phages impact methane oxidation to a significant degree. In comparison, the ratio of phage sulfur AMGs identified by us is 0.00016% which is one magnitude higher than that observed for methane oxidation. Therefore, we acknowledge that while these AMGs are rare, their impacts should not be discounted.

- Although AMGs may be involved in biogeochemical cycles, viruses do not need these enzymes to have important environmental impacts. Arguably the major influence of bacteriophage on elemental cycles is their killing of host cells. Other phage that do not encode AMGs could therefore have a much bigger impact on sulfur cycling than those described here. The AMGs are potentially only active for short periods of time during a brief window of the infection, so gene or transcript abundances may give a misleading view of their importance, especially for isolated metagenomes/transcriptomes. I encourage the authors to include these caveats in their study. Right now it reads a bit like AMGs are the only way phages can shape elemental cycles.

- **Reply:** We agree with the reviewer that AMGs are not the only mechanisms for phages to shape biogeochemical cycles. We provide the following sentences in the manuscript to detail that lysis represents a significant component of phage impacts on biogeochemistry: “Despite their reduced genome size and limited coding capacity, phages are known for their ability to modulate host cells during infection, take over cellular metabolic processes and proliferate through a bacterial population, **typically through lysis** of host cells” ... “As many as 20-40% of all bacteria are assumed to be in a virocell state, undergoing phage-directed metabolism” ... “**One such mechanism** by which phages can alter the metabolic state of their host is through the activity of phage-encoded auxiliary metabolic genes (AMGs)”.

We have made an effort for it to be clear that “**typically through lysis**” is the main mechanism of host manipulation and that AMGs represent only “**one such mechanism**”. To address this concern we have added further clarification of this caveat in the discussion section (line 509).

- Little information on the hosts infected by these sulfur phages is given, which limits the ability to infer any potential biogeochemical impact. Understanding the host is critical to evaluating the environmental role of the virus, but that is not addressed in this study. Recent work examining methane metabolism AMGs in freshwater phage was able to reconstruct complete genomes of novel phage and provided in-depth links to predicted hosts, for example, so a more complete picture of how phage influence biogeochemical cycles was provided (Chen, 2020, Nature Microbiology PMID: 32839536). To improve this study the authors could perform CRISPR or co-occurrence analyses of the phage they found to identify putative hosts. Given the diverse hosts involved in sulfur cycling it is likely that the AMGs have different roles depending on the host, and this possibility should be discussed.

- **Reply:** In our experience, CRISPR spacer analysis does not always identify a host organism. It is assumed that only ~7% of phage hosts can be identified according to this method (Paez-Espino et. al., 2016, NAR). Furthermore, there is little evidence for co-occurrence to yield accurate phage:host pairs. In response to this suggestion, we have performed CRISPR analysis by identifying CRISPR spacers from 25 metagenomes that included the phages presented in this study (line 304, 588). We identified no host matches.

Instead, we reconstructed phylogenetic trees to find hosts for phages in hydrothermal environments, freshwater lakes, marine environments (*Tara* Ocean datasets), and coastal environments (Chesapeake Bay) (Supplementary Figures 8-11). Since phages acquire AMGs from recombination of a respective host gene, phylogenetic relatedness can be used to estimate phage:host pairs.

Furthermore, we include the following excerpts in the manuscript to highlight how “AMGs have different roles depending on the host”. These statements assume that individual host metabolisms affect the role(s) of the AMGs.

- Line 291: “On the basis of these findings, we hypothesize that the function(s) of DSM AMGs during infection is most likely constrained by specific host sulfur metabolisms rather than viral taxonomy.”
- Line 318: “A lack of universally shared protein groups likewise suggests the DSM AMGs function independently of other host metabolic pathways and likely strictly serve to supplement host DSM pathways.”
- Line 440: “First, our findings support previous hypotheses that viral metabolism targets key or bottleneck steps in host metabolic pathways.”
- Line 463: “Taken together, these observations support the conclusion that viral auxiliary metabolism targets key steps in host metabolic pathways for finely tuned manipulation of energy production or nutrient acquisition.”

Figures 2 and 8 are very general and summarize broad aspects of phage infection and elemental cycling that are already known. In general the manuscript could be shorted to focus on the novel aspects of this study (detailed phylogenies on all marker genes identified) and less on that which has already been reported by other studies.

- **Reply:** We provide detailed phylogenies of all AMGs and respective bacterial homologs (Figure 3 and Supplementary Figures 8-11). Figures 2 and 8 serve as important conceptual diagrams of how AMGs function in host cells and the overall community. Since these two concepts are the central focus of this manuscript we have elected to retain them as main text figures. We likewise believe these figures are useful to readers less familiar with phage manipulation of metabolism.

Groups such as podoviridae, myoviridae, and siphoviridae are defined primarily by morphology, so family level classifications cannot be accurately made based on genomic data alone. The classifications of many phage in NCBI are incorrect, so these errors will be propagated here. Many of these families appear mixed together in Figure 4, and it is unclear if any of the data is supported by actual microscopy.

- **Reply:** Genomic data, specifically the encoded proteins, provides sufficient information to classify phages by family, especially for *Caudovirales* (tailed phages). The accuracy of this methodology, in terms of protein sharing networks, has been published previously (Jang, 2019, Nature Biotechnology, PMID: 31061483). In addition, the methodology for protein

sharing to reference databases has also been previously used in a paper provided by the reviewer (Al-Shayeb, 2020 Nature, PMID: 32051592).

Moreover, we believe the claim that many “phage classifications in NCBI are incorrect” is an anecdotal statement. Phage taxonomy classifications on public databases such as NCBI RefSeq are in fact backed by microscopy imaging. Microscopy is not an option when analyzing metagenomic phages. Again, no studies provided by the reviewer perform microscopy but they provide taxonomic assignments based on protein sharing networks of phages and reference database similarity. Based on evidence in the reference manuscripts provided we believe the taxonomic assignments presented here are accurate.

The network figure is general and difficult to interpret, especially given the problems with viral taxonomy and incomplete genomes. It is difficult to interpret the finding that sulfur phage cluster with known phage of pelagibacter and cyanobacteria. It seems like this is artefact. One can only conclude that the sulfur phage sometimes cluster with known phage. A general summary figure showing how many contigs were similar to known phage would be more useful.

- **Reply:** To address this comment we have added clarification that there is not significant enough protein similarity between the phages presented and reference phages appearing in the network figure to accurately identify host range (lines 295-297, 301-305). We clarified that the network figure indicates taxonomy rather than host range. We furthermore removed the phrase “Most vMAGs clustered with reference phages that infect *Pelagibacter*, Cyanobacteria and Enterobacteria”.

The section titled “Estimates of sulfur phage contributions to sulfur oxidation” should be renamed since no actual measurements of sulfur oxidation have been made. This section appears to be based on gene abundance ratios, which are very different.

- **Reply:** We changed the title of this section as “Estimations of sulfur phage contributions to sulfur oxidation based on omics-data analyses”.

Reviewer #2 (Remarks to the Author):

In this manuscript, Kieft, Zhou, et al. study the ecology, metabolism, and distribution of phages that infect sulfur species and carry AMGs involved in sulfur metabolism. The authors take advantage of a dataset consisting of 191 phage scaffolds reconstructed from metagenomes that contain phage AMGs related to sulfur metabolism. The authors identify two previously unknown sulfur-related AMGs and present multiple (and, to the best of my knowledge, novel) analyses that put sulfur AMGs in environmental, ecological, and host-phage interaction contexts. These include identifying the potential role of sulfur AMGs in virocell metabolism, evaluating the abundance of sulfur phages and sulfur AMGs across different environments, determining sulfur phage taxonomy and host range, and more. Overall the manuscript is well written. The analyses presented are insightful and of high quality, and the authors make good use of public data. The paper could be of interest to researchers interested in microbiology, sulfur metabolism, and biogeochemical cycles (sulfur in particular, of course).

Reply: We thank the reviewer for their positive comments.

- “As many as 30-40% of all bacteria are assumed to be in a virocell state, undergoing phage-directed metabolism” (lines 69-70): in Curtis Suttle’s paper, it is stated that “viral lysis in surface waters removes 20–40% of the standing stock of prokaryotes each day”. Lysing is different from being in a virocell state. If 20-40% of the cells are lysed every day, I would assume that the share of virocells from the total prokaryotic cells is significantly lower, depending on the estimated average phage latent period. The share of virocells is important because it is used later (456-460).

- **Reply:** We believe this comment to be based on confusion that we can clarify here. A virocell, by definition, is an infected cell. Therefore, in order to be lysed by a virus, a cell must have been in a virocell state. The 20-40% of prokaryotic cells lysed by viruses should directly translate to those 20-40% of cells having been in a virocell state at any given time. Given that not all infected cells lyse and there is latency before lysis, the proportion of cells within a virocell state would be greater than those lysed, not less.

- “We identified 190 viral metagenome-assembled genomes (vMAGs) and one viral single-amplified genome” (lines 123-124): The use of the term “vMAGs” is a bit of a stretch in this case, “viral scaffolds” may be more appropriate. The majority of scaffolds (108/191) are shorter than 20Kbp, which is roughly half the size of a podovirus genome. Considering that most of the scaffolds are predicted to represent myovirus genomes, which are much longer, the majority of scaffolds are probably far from representing genomes.

- **Reply:** We have added a sentence to be clear that the vMAGs are likely partial genome scaffolds (Line 135). Moreover, we have replaced the term “vMAG” with “mVC” (metagenomic viral contig). The term “mVC” is consistent with the nomenclature presented in Paez-Espino et al., 2017 (source of the IMG/VR database) and better explains that the scaffolds are partial rather than complete genomes.

- “Most vMAGs clustered with reference phages that infect *Pelagibacter*, Cyanobacteria and Enterobacteria ...” (283-287): it makes sense to me to include the information available in Supp. Fig. S8-S11 when discussing the potential hosts of the sulfur phages. I assume the information in the phylogenetic trees is more reliable for this purpose.

- **Reply:** As stated above we removed the phrase “Most vMAGs clustered with reference phages that infect *Pelagibacter*, Cyanobacteria and Enterobacteria” and clarified that the network diagram is best utilized as an analysis of taxonomy. We have added a sentence (line 296) to clarify that although vMAGs (mVCs) clustered with the indicated reference phages there was limited protein similarity. We have added further information regarding the host ranges (referencing Figure 3) and results from CRISPR analysis.

- “Only a few protein groups were globally shared amongst the vMAGs, including common phage proteins” (294-295): Could this be the result of the partial genomes available for the phages? 132 vMAGs were identified as myoviruses, which typically have long genomes, at the range of >100Kbp. However, there are only 9 vMAGs longer than 100Kbp. Since the dataset consists only of scaffolds with sulfur AMGs it is likely that many other phage genes are missing.

- **Reply:** We have added a sentence to be clear that a caveat of the protein sharing diagram is that a significant number of vMAG (mVC) proteins are likely missing from analysis since the vMAGs (mVCs) are partial scaffolds (line 314). However, we do not anticipate that this affected the results of the protein grouping in any significant manner since phages

rarely share any large portion of proteins. This is highlighted by the alignments presented in Figure 5 b-d showing that similar phages, by genome alignment, cluster together on the protein grouping plot.

- “Conservatively assuming that 40% of all sulfur-oxidizing SUP05 Gammaproteobacteria are infected by sulfur phages (in line with observations of phage infections in the pelagic oceans) ...” (lines 457-460): see above comment regarding the 40% figure. Also: sulfur-oxidizing SUP05 Gammaproteobacteria may be infected by phages that do not contain sulfur AMGs and therefore do not affect the host’s sulfur metabolism. Therefore, the energy estimate seems too high (or the assumptions need to be explained in more detail).

- **Reply:** In Fig. 6a, b, we have calculated the vMAG (mVC) AMG to total gene ratios (here, total genes include phage AMGs and putative bacterial host genes within the whole community). This ratio reflects the sulfur phage *dsrA* abundance within the whole community. The scenario of some phages not containing sulfur AMGs would not influence this ratio. As shown in Fig. 6a, b, compared to Fig. 7a, b, the average phage:total gene coverage ratios within the whole community are lower than those of individual phage-host pairs (Tara Ocean samples: 34% vs 93%; Hydrothermal samples, Lau Basin: 7% vs 46% and 61%; and Freshwater lakes: 3.1% vs 73%).

As suggested by the reviewer, we estimate that 20-40% of all sulfur-oxidizing SUP05 Gammaproteobacteria are infected by sulfur phages in marine settings. We then estimated phage:total gene coverage ratio within the whole community by multiplying virocell percentage with the average phage:host gene coverage ratio within phage-host pairs (as the phage gene coverage), and then divided by total gene coverage. While estimated ratios are not consistent for the observed ratios (hydrothermal environment: estimated ratio 15-38% vs observed ratio 7%), we believe that, as the reviewer suggests, this could result from that a considerable fraction of host cells that are infected by phages without DSM AMGs, and/or result from lower percentage of virocells than assumed by us. We added this information as a caveat to the main context to point out the differences between these two sets of numbers and provide the biological explanation to this phenomenon (lines 383-394).

We agree the energy estimate is definitely high if we count 40% of sulfur-oxidizing SUP05 Gammaproteobacteria virocells all contain the viruses with DSM AMGs (specifically, sulfur-oxidizing AMGs here). We have now changed this accordingly as suggested (line 506). Now the statement reads: “Assuming that in Guaymas Basin, phage:total *dsrA* gene coverage ratio is 10% (as the average level in sampled hydrothermal environments (such as Lau Basin), it may be estimated that 282.2 J/Kg of energy may in fact be transformed by sulfur AMGs.”

Minor comments:

- Fig. 1 (also possible in the text): how does the organization of the sulfur genes on the phage genomes compare to their organization on the genomes of their potential hosts?

- **Reply:** Phage and host genome arrangements are too divergent to be comparable. Phages have approximately 3 genomic regions corresponding to auxiliary, nucleotide replication and structural assembly cassettes, whereas host genomes are vastly more complex. Host genomes encode metabolic genes, specifically those for dissimilatory sulfur oxidation, in operons or gene clusters. This is not a common theme among phages which encode typically 1-2 genes of a metabolic pathway.. Although we acknowledge that this question

may be posed by other readers we have elected to omit this information from the manuscript as it provides no useful insights.

- Fig. 6b, 7b: TARA_B100000034 appears twice, is this intentional?

- **Reply:** There are two metagenomes under this sample ID as stated in Supplementary Table 4. I added a statement in the caption of Fig. 6b, 7b to clarify this.

- Sentence in lines 453-456 may be incomplete?

- **Reply:** We corrected the grammar mistake in this sentence.

Reviewer #3 (Remarks to the Author):

Kieft and Zhou et al. have followed up on earlier work identifying sulfur-cycling genes in phages to demonstrate how widespread this phenomenon is. There appears to be an higher representation of these genes in phages in certain environments than one would expect by pure chance. These metabolic genes are highly expressed, and a phage introducing extra gene copies would have an impact on the final amount of protein produced for rate-limiting steps and provide a fitness advantage for the organism. It should be noted that this and earlier work in this field is still somewhat speculative - there doesn't seem to have been any experiments confirming this supposition in laboratory cultures. However, the quantitative evidence from metagenomes (in particular figures 6 and 7) is hard to argue with and it seems that the authors are really onto something here.

Reply: We thank the reviewer for their positive comments.

The only major criticism that I have with this work is that the authors examine a lot of different environments without taking into account whether or not there is any sulfur cycling (sulfate reduction/sulfide oxidation) there. One would expect very little sulfur-cycling in fully oxygenated aquatic environments, for example. Which of the Tara Ocean samples were from OMZs, for example? Wouldn't one expect anoxic environments with more selection for sulfate reduction to have higher phage:host ratios? So a more nuanced discussion of the geochemical conditions in the environments where this data (especially quantitative data in figures 6 and 7) comes from could be helpful, with any quantitative patterns relating to the predicted degree of sulfur cycling highlighted.

- **Reply:** These *Tara* Ocean samples are all from epipelagic zones, thus are not located in the OMZ. Phage AMGs discovered from these *Tara* Ocean samples indicate that sulfur oxidation processes do exist and function in this environment. A recent publication shows that there are abundant sulfur-oxidizing bacterial taxa (including SUP05) in both particle-associated and free-living fractions across the oxic–anoxic interface, with peak abundance at the shallow anoxic layer, which indicates there are still considerable amount of sulfur-oxidation processes happening in marine oxygenated zones (Suter et al., 2017, *Environmental Microbiology*, 10.1111/1462-2920.13997). Additionally, the first cultivated SUP05 strain, *Ca. Thioglobus* autotrophicus strain EF1, was also isolated from suboxic zone and can facultatively grow under aerobic condition (Shah et al., 2017, The

ISME Journal, 10.1038/ismej.2016.87). Anoxic environments would be ideal for sulfate reduction to take place, but we believe the reviewer may have misinterpreted this. Our study focuses solely on oxidative bacterial processes (sulfur and thiosulfate oxidation) and phage AMGs (*dsrA*, *dsrC/tusE*, *soxC*, *soxD*, and *soxYZ*) mediating these processes.

The environments from which we have used the samples in this study do not have very detailed geochemical characterization. As suggested by the reviewer, we now provide additional qualitative information to help the readers to have a better general understanding of these environments (Section at line 344).

- line 69-70: "of all bacteria" is a very broad statement - this should be qualified to be limited to the environments investigated in the citations 10 and 11.

- **Reply:** We have amended this accordingly to read that this statement is "according to some estimates" and refers to "aquatic environments".

- lines 73-77: More of an explanation for how these AMGs work would be helpful here. It is critical that these genes are highly expressed central metabolic genes encoding rate-limiting steps - otherwise adding an additional copy on the phage genome would not provide any fitness advantage. The other lingering question is why hasn't evolution simply boosted the copy number on the host's genome? This suggests that phage infection only boosts fitness sometimes. A more in-depth discussion of these nuances would help a reader like me coming from outside the phage AMG field trying to understand how it all fits together.

- **Reply:** The additional copy of the metabolic gene is advantageous for phage infection of the host and the host does not benefit (rather is lysed). Adding an extra copy of the metabolic gene to the host genome would only occur under positive selective pressures on the host, which do not exist because the host is lysed by the phage (negative pressure). The positive selective pressures to acquire additional copies of the gene are on the phage genome, and therefore the extra copy would appear on the phage, not the host. It has been hypothesized that AMGs function to provide a fitness advantage to phages only under specific conditions for driving specific phage-infection-related processes. In situations by which the host benefits from additional copies of a metabolic gene we typically observe gene duplication within the host genome. There are several examples for which bacteria encode multiple copies of *dsrC* ([10.1038/s41396-018-0078-0](https://doi.org/10.1038/s41396-018-0078-0), [10.1016/j.bbabi.2014.03.007](https://doi.org/10.1016/j.bbabi.2014.03.007)). We have added further background information and clarification of this topic (line 78-79, 85, 89-92).

- line 96: "and" should not be italicized

- **Reply:** We have corrected this in the main text.

- line 128: this statement needs a citation.

- **Reply:** Citations for the identification of *dsrA*, *dsrC* and *soxYZ* genes on phages were provided within the Introduction.

- Figure 2: The way these figures are drawn it seems that phage infection takes the bacteria from having no DsrAC or SoxYZ/CD to having it, whereas as I understand it the model is that the phage genes augment the genes with the same function that the hosts already have? It would be clearer if

the figures were modified to show that this is an augmentation rather than an initial introduction of the genes.

- **Reply:** We have added clarification in the figure legend to be explicit that microbial (host) pathway enzymes and sulfur carriers are active alongside the phage proteins.

- Figure 2: The chemical reactions in the boxes are confusing - the "complete reactions" are half reactions and the "reaction catalyzed by" need other enzymes than those mentioned. Chemical stoichiometry is not relevant to the story so they could easily be removed without detriment to the manuscript.

- **Reply:** As suggested we have removed the chemical reactions to eliminate confusion.

- Figure 6: Why aren't all genes being shown for all environments? Are the ones not shown just 0% (no *dsrA* or *soxYZ* on any phages at all)?

- **Reply:** Yes, not all the phage-encoding *dsrA* and *soxYZ* genes were found to be present in all environments. The ones that are not shown in the figure are just not present in the environment, which means their abundances are just 0%. We have counted and listed all phage AMGs for each environment.

- Line 322: "Estimates of sulfur phage contributions to sulfur oxidation" - this title seems to over-promise a bit. Wouldn't the fraction of infected cells be necessary knowledge to truly estimate the contribution to sulfur oxidation? That is not data that is available in this study, isn't it?

- **Reply:** As pointed in the above comments by the reviewer, the fraction of infected cells is not enough to tell the contribution to sulfur oxidation by sulfur phages due to that some infections are by phages without having the sulfur oxidation AMGs. While the point raised also leads to another consideration we might have neglected that sulfur phage gene ratios do not necessarily and sufficiently reflect the real situation of the phage contribution to sulfur oxidation. Other possibilities still exist. For example, there might be different transcriptional activities between phage and host genes. The real activities of virocells and non-virocells could be different due to their different biology that virocells might have higher or lower environmental fitness so that they are more or less active in mediating the sulfur oxidizing processes. We admit that our statement in the context of this section needs to be reorganized to avoid addressing the issue in a simple and direct manner. We placed our statement in a more concise and detailed qualification as suggested. As the section was based on omic-data, we also changed the title to "Estimations on sulfur phage contributions to sulfur oxidation based on omics-data analyses".

- lines 382-392: There are several places in this paragraph where genes are referred to, e.g. line 385 "phage to host *dsrA* gene ratios", when it seems the authors mean to refer to gene transcripts?

- **Reply:** We changed them to "transcript" in these places.

- line 387: So infinity means that there was zero host gene coverage? Maybe just say that?

- **Reply:** Yes, we added this information in this sentence. Now, it is "phage to host *dsrA* transcript ratios varied from 1.9 to infinity (host transcript abundance is zero)." (line 423).

- Figure 7AB: Why are these data shown as ratios, while they are shown as percentages in figure 6? Shouldn't this be consistent throughout the paper?

- **Reply:** In Fig. 6, it is the percentage of phage:total, while in Fig. 7a, b, it is the ratio of phage:host. In Fig. 7a, b, because that phage gene abundance can be several times higher than that of host gene abundance, if we use percentage, that will be multiple times higher than 100%, which is not a very appropriate expression. We used percentage to reflect the phage:total ratios in both Fig. 6 and Fig. 7, which is consistent throughout the paper.

- line 411: Why is it likely that DsrA has a greater affect than DsrB? This statement needs some degree of justification.

- **Reply:** We have removed this sentence as there is no evidence to support this hypothesis.

- line 417: "is provide" should be "is provided"

- **Reply:** We have fixed this accordingly.

- line 458: More details for this calculation is needed - what numbers are being used for cell numbers and rates here?

- **Reply:** We made changes accordingly to make the details for this calculation clear (lines 386-389).

- line 477 and figure 8b: It's important to keep in mind that a lot of biomass created in these environments will end up buried in the deep subsurface and turned over very slowly, rather than entering the food chain, thus contributing to global controls on oxygen and CO₂. Could these phages be helping to regulate the global climate? That might be an important point to include in the discussion and figure 8.

- **Reply:** As suggested by the reviewer, the biomass that is being created in the reduced sulfur oxidizing process could be buried in the deep subsurface, thus not being included into the food chain and open biosphere. We added that into our discussion (lines 534-538). In Fig. 8b, we also added sedimentation of POM as one of the fates of biomasses created in these environments.

- line 494-495: Please specify how the IMG/VR database was queried for dsr and sox. Was this HMMer? BLAST? Text-based search? KEGG? COG? What was it?

- **Reply:** We have added clarification that this was an HMM query of existing annotations on the database (line 549).

- line 497: Does this mean that all 192 passed screening with these tools, or was there a larger number and some were discarded?

- **Reply:** We have added further details of the methods that were accidentally left out (Methods line 547). This included information about how the AMGs were originally pulled from IMG/VR. This method was the use of custom HMMs (citation provided in text) and associated trusted cutoff scores.

- line 504: "not all vMAGs were predicted as phage" - does this refer to the five vMAGs talked about on line 501? Or others?

- **Reply:** We have added clarification that this refers to all of the 192 vMAGs (mVCs) discussed to that point (line 559).

- line 540: "gaps were stripped by 90% (SoxD and SoxYZ) or 98% (DsrA and DsrC/TusE)" - is there another way of expressing this? It seems unclear what was done with the gaps.

- **Reply:** This term "gaps stripped" is a standard term for eliminating those columns from the alignment that contained X% gaps. We have added "(masked)" after the term "stripped" to try and add further clarity (line 609).

REVIEWER COMMENTS

Reviewer #2 (Remarks to the Author):

The revised manuscript addresses most of my concerns. However, two issues are still not fully addressed:

- "As many as 30-40% of all bacteria are assumed to be in a virocell state, undergoing phage directed metabolism" (lines 69-70)...": I may be missing something here; my apologies if I do. I agree that it is reasonable to estimate that 30-40% of the cells are in a virocell state EVERY DAY, but this does not translate to 30-40% at ANY GIVEN TIME. To know what fraction of the cells are virocells at any given time, we need to know the average time a cell spends in the virocell state before it is lysed (or "recovers"). If the latency time is one day, then on average, 30-40% of the cells are virocells at any given time. However, if the latency time is, say, one hour, then at any given time $\sim 1-1.5\%$ of the cells are in virocell state (because, on average, $\sim 1-1.5\%$ of the cells are lysed every hour to add up to 30-40% every day). To know what the correct fraction is, we need to have an estimate of the latency time. It is important to state the correct figure to have a reliable estimation as to the contribution of phage metabolism to marine bacterial metabolism. Also, this will affect the energy calculations provided in the text.

- "Only a few protein groups were globally shared amongst the vMAGs, including common phage proteins"... - I think that the added sentence does not solve the problem. To my understanding, the fact that this analysis results are consistent with the taxonomic clustering has little meaning if the true reason for the lack of universal families is data partiality. In such a case, even if there are many universal families in reality, it is still very possible that the authors would get the results that they got. I suggest that the authors would do one of the following to make this analysis more meaningful:

1. Create a model that predicts the number of universal protein families given the observed data
2. Focus on neighboring genes to the sulfur genes that were used to identify the vMAGs (per-gene analysis). These sets of genes are probably more complete. Therefore high diversity in these groups can be claimed to demonstrate high genomic variation.
3. Provide some other analysis that will show that the observed diversity is due to high genomic diversity.

- Following one of the comments made by Reviewer 1, I would like to ask the authors to add an estimation of the portion of the time in which a virocell is expected to exhibit phage metabolism. This is necessary for estimating the contribution of phage metabolism to global metabolism and energy production.

Written by Itai Sharon

Reviewer #3 (Remarks to the Author):

My concerns with the manuscript have all been addressed. I'm not sure that the correct version of figure 2 was uploaded so the authors may want to double check that.

Responses to responses to reviewer 3:

"The environments from which we have used the samples in this study do not have very detailed geochemical characterization. As suggested by the reviewer, we now provide additional qualitative information to help the readers to have a better general understanding of these environments (Section at line 344)."

This extra geochemical context is helpful.

"line 69-70: "of all bacteria" is a very broad statement - this should be qualified to be limited to the environments investigated in the citations 10 and 11.

- Reply: We have amended this accordingly to read that this statement is "according to some estimates" and refers to "aquatic environments".

This statement more accurately reflects the citation now.

"We have added further background information and clarification of this topic (line 78-79, 85, 89-92)."

This additional background information is welcome.

"Citations for the identification of *dsrA*, *dsrC* and *soxYZ* genes on phages were provided within the Introduction."

There is no problem in repeating these citations to help the reader follow the authors' arguments, but it's true that it's not a major problem the way it's currently written.

"We have added clarification in the figure legend to be explicit that microbial (host) pathway enzymes and sulfur carriers are active alongside the phage proteins."

This is helpful.

"As suggested we have removed the chemical reactions to eliminate confusion."

They appear to still be there in the version of figure 2 that I am seeing in the revision?

"As the section was based on omic-data, we also changed the title to "Estimations on sulfur phage contributions to sulfur oxidation based on omics-data analyses"."

This is a better heading.

"In Fig. 6, it is the percentage of phage:total, while in Fig. 7a, b, it is the ratio of phage:host. In Fig. 7a, b, because that phage gene abundance can be several times higher than that of host gene abundance, if we use percentage, that will be multiple times higher than 100%, which is not a very appropriate expression. We used percentage to reflect the phage:total ratios in both Fig. 6 and Fig. 7, which is consistent throughout the paper."

This distinction makes sense, my mistake.

"line 458: More details for this calculation is needed - what numbers are being used for cell numbers and rates here?"

- Reply: We made changes accordingly to make the details for this calculation clear (lines 386-389)."

I cannot find the exact same numbers as in the original version of the manuscript but there appears to be some better supported calculations resulting in a different answer, so I guess that's a positive improvement to the manuscript.

"As suggested by the reviewer, the biomass that is being created in the reduced sulfur oxidizing process could be buried in the deep subsurface, thus not being included into the food chain and open biosphere. We added that into our discussion (lines 534- 538). In Fig. 8b, we also added sedimentation of POM as one of the fates of biomasses created in these environments."

Great - I like this added discussion point.

Comments about the methods section have been adequately addressed.

Responses to responses to reviewer 1:

"

•

Kieft et al present an analysis of sulfur auxiliary metabolic genes (AMGs) encoded by bacteriophage present in the IMG/VR database. The authors annotate genes belonging to the *dsrA*, *dsrC/tusE* and *soxCDYZ* families, and provide some gene phylogenies and genomic comparisons of the phage sequences that encode these genes. They also analyze some metatranscriptomes to show that these genes are expressed in some cases.

The authors do not present new metagenomes, and this study is an analysis of sequences available in the IMG/VR database. It is good that researchers use IMG/VR for these purposes, but the novelty of this study is limited since this database has been available for years.

Reply: We disagree with the premise that old datasets do not contain novel information. Within his/her review, the reviewer has cited Al-Shayeb et al., 2020 Nature (PMID: 32051592) as a model paper for describing a 700kb phage. This manuscript utilized both the Tara Oceans and Global Ocean Virome databases which are also years old (2011 and 2016, respectively). These databases (IMG/VR, Tara Oceans, GOV) are constructed as repositories of data for the discovery of new and useful information. The recent manuscripts "A new view of the tree of life" (Hug et. al., 2016, PMID: 27572647, Nature Microbiology) as well as "A genomic catalog of Earth's microbiomes" (Nayfach et. al., 2020, PMID: 33169036, Nature Biotechnology) likewise utilized publicly available data (e.g., IMG/M) and are widely accepted in the field. To use genomes from publicly available databases for addressing scientific questions should not be considered as lacking novelty. Novelty roots in how to interpret biological and ecological outcomes

from datasets and whether the results are scientifically remarkable and insightful. Since the rapid increase of data sizes and development of new methods, many scientists have leveraged growing-size of datasets to conduct novel and insightful research."

I agree with the authors here - genomic and metagenomic datasets are large and complex and every possible question cannot be addressed with a new dataset in its initial publication. Re-analysis of existing datasets with new questions should be encouraged if we want to make the most out of our data.

"No novel approaches for analyzing the existing data are presented.

•

Reply: Multiple studies discussed by the reviewer (Chen et al., 2020, Al-Shayeb et al., 2020) to refute the impact of the data presented here contain no novel approaches. Beyond CRISPR spacer analysis to predict hosts, which we have now implemented, the two studies follow similar methodology for studying phages encoding AMGs (transcriptomics, AMG phylogeny, phage phylogeny, environmental nutrient context, conceptualization of AMG impact, etc.). The reasoning for this is straightforward: the methodology used in these two articles, in addition to the methods displayed in the data presented here, reflect the most state-of-the-art methods for metagenomic phage analysis. The reviewer has expected that this manuscript presents both novel information (i.e., expanding known DSM AMGs on phages) as well as generates new methodology, which was not the scope of the study."

I agree with the authors here - the methods and the data aren't necessarily novel, but the framing, the questions and the hypotheses are new. This is novel combination of methods and data that has never been done before, and that is where the novelty lies.

"The 190 metagenome-assembled viruses appear to actually be viral contigs, not complete genomes, and some are quite short (5 kbp) and the average is only 31 kbp...."

The authors have responded well to these concerns - the phage genomes analysed, while not complete, are not substantially smaller than publically available genomes in NCBI. The authors have also done all that they can to verify that these are indeed phage sequences. It's true what the reviewer says - that much more could be done with complete phage genomes - but that doesn't make the conclusions of this study less valid.

"

- Reply:

The 190 viral contigs of sulfur phages comprise a small proportion of IMGVR, which contains hundreds of thousands of contigs. So less than 0.01% of the contigs had sulfur metabolism genes- this statistic should be calculated exactly and provided to put these findings in context. It seems that these AMGs are rare and found in specific environments, so the claims of the importance of sulfur phage for global elemental cycles are overstated and should be qualified.

The small percentage of 'rare' phage *dsrA* genes in the database does not underestimate the importance of sulfur-oxidizing bacteria or phages with DSM AMGs in the global biogeochemical cycling process. To highlight this we analyzed the abundance of sulfur oxidizing bacteria in the IMG/M database (microbial equivalent to IMG/VR). We searched ~56 million bacterial scaffolds of which only ~1000 scaffolds encoded *dsrA* genes (0.0018%). Despite this small percentage, bacteria encoding *dsrA* genes are known to significantly contribute to global sulfur biogeochemistry and cycling.

In a recently published paper (Chen, 2020, Nature Microbiology PMID: 32839536), viruses containing methane oxidizing AMGs were found to occupy a low portion in the total community (0.0000218%). Despite this, it is hypothesized that these phages impact methane oxidation to a significant degree. In comparison, the ratio of phage sulfur AMGs identified by us is 0.00016% which is one magnitude higher than that observed for methane oxidation. Therefore, we acknowledge that while these AMGs are rare, their impacts should not be discounted."

I agree with the authors here - the fact that there are very few phage genomes containing the relevant genes - this says nothing about their abundance in the environment as the database is not an accurate representation of the environment. As the authors point out, all the biogenic sulfide in the world is made by sulfate-reducing bacteria making up a vanishingly small fraction of all metagenomes and yet nobody claims that their effect on biogeochemical cycling is insignificant.

"Although AMGs may be involved in biogeochemical cycles, viruses do not need these enzymes to have important environmental impacts. Arguably the major influence of bacteriophage on elemental cycles is their killing of host cells..."

The authors responded well to this criticism - AMGs certainly aren't the only way that phages affect biogeochemical cycles and their addition of statements recognising this fact is a good response.

"Groups such as podoviridae, myoviridae, and siphoviridae are defined primarily by morphology, so family level classifications cannot be accurately made based on genomic data alone..."

Here the authors appear to be doing the best they can to classify the phage based on metagenomic sequencing and cite papers by others doing the same. It's true that the NCBI taxonomy database isn't perfect, but there is no better alternative for this kind of study. Since some taxonomic inaccuracies would not change any of the conclusions of this manuscript I don't think that this is a problem, and as they clearly state their methods then there is no attempt to hide these potential inaccuracies from informed readers.

"The network figure is general and difficult to interpret, especially given the problems with viral taxonomy and incomplete genomes..."

The extra caveats that the authors have added to the interpretation of the network figure help.

Responses to the second round of reviewer comments

Reviewer #3

I'm not sure that the correct version of figure 2 was uploaded so the authors may want to double check that.

Authors' Reply:

We mistakenly did not upload the correct version of Figure 2. We have now included the updated figure which does not include reaction diagrams.

Reviewer #2

- "As many as 30-40% of all bacteria are assumed to be in a virocell state, undergoing phage directed metabolism" (lines 69-70)...": I may be missing something here; my apologies if I do. I agree that it is reasonable to estimate that 30-40% of the cells are in a virocell state EVERY DAY, but this does not translate to 30-40% at ANY GIVEN TIME. To know what fraction of the cells are virocells at any given time, we need to know the average time a cell spends in the virocell state before it is lysed (or "recovers"). If the latency time is one day, then on average, 30-40% of the cells are virocells at any given time. However, if the latency time is, say, one hour, then at any given time ~1-1.5% of the cells are in virocell state (because, on average, ~1-1.5% of the cells are lysed every hour to add up to 30-40% every day). To know what the correct fraction is, we need to have an estimate of the latency time. It is important to state the correct figure to have a reliable estimation as to the contribution of phage metabolism to marine bacterial metabolism. Also, this will affect the energy calculations provided in the text.

Following one of the comments made by Reviewer 1, I would like to ask the authors to add an estimation of the portion of the time in which a virocell is expected to exhibit phage metabolism. This is necessary for estimating the contribution of phage metabolism to global metabolism and energy production.

Authors' Reply:

As suggested by Reviewer #2, we looked into the detailed description of percentage of infected bacterial cells (virocell percentage).

However, we would like to state that we did not use the virocell percentages for calculation of thermodynamic estimates for energy conversions from AMGs. Instead, we used observed phage:total gene ratios at the community level for these estimations. While this does not take into account the percentage of virocells, or burst sizes of infected cells, we think it provides the most reasonable estimate for energy conversions driven by phage AMGs. We want to clarify that these virocell percentages are only provided for reference for the viewers.

We agree that bacterial mortality rates and % of cells in virocells can be decoupled from each other - we had not recognized this previously. To calculate the percentage of virus genes within the total community, we need to know the virocell percentage at any given time which represents virus-infected bacterial cell ratios within the community, but not the bacterial mortality rate. In order to find the representative percentage of infected bacterial cells in different environments (mainly marine and freshwater environments), we looked into the details described in references below (including the percentage ranges and methods to obtain these percentages).

In Ref #1:

Quote: "... from 3 to 31% of the free-living bacteria and 3 to 26% of particulate-associated bacteria appeared to be phage-infected at any given time" (*Microbial Ecology* 25.2 (1993): 161-182). Ref #1 used a one-step growth experiment including all timepoints of the phage latency period to identify associations between visibly infected bacteria to all virus-infected bacteria. This data was used to calculate the range of infected bacterial cells at any given time in ocean systems.

In Ref #2:

Quote: "We estimated that 1 to 17% \pm 12% of all bacteria were phage infected, implying that phage-induced mortality was <34% \pm 24% of total mortality." (*Applied and Environmental Microbiology* 61.1 (1995): 333-340). Ref #2 calculated the percentage of phage infected bacteria in freshwater lakes using the same TEM method described in Ref #1.

Based on these two references, we use the mean values to estimate virocell percentages at any given time for marine and freshwater environments: 1) 16% virocell percentage for marine bacteria (the mean value for ranges of 3 to 31% and 3 to 26% in free-living and particulate-associated marine bacteria), 2) 15% virocell percentage for freshwater bacteria (the mean value for range of 1 to 29% in freshwater bacteria). These estimated virocell percentages were subsequently used to perform the comparison between estimated phage:total gene coverage ratio within the whole community with observed ratio in our main text (Line 397).

This virocell percentage cannot be easily translated to bacterial mortality rate, but could be converted to phage-induced bacterial mortality provided that the latency period matches the generation time of uninfected bacteria. Using this, phage-induced mortality may be calculated as growth rate of bacteria \times fraction of infected bacteria. The local bacterial growth rate and latency time of phages should both be taken into consideration if we want to convert bacterial mortality rate to virocell percentage.

- "Only a few protein groups were globally shared amongst the vMAGs, including common phage proteins"... - I think that the added sentence does not solve the problem. to my understanding, the fact that this analysis results are consistent with the taxonomic clustering has little meaning if the true reason for the lack of universal families is data partiality. In such a case, even if there are many universal families in reality, it is still very possible that the authors would get the results that

they got. I suggest that the authors would do one of the following to make this analysis more meaningful:

- 1. Create a model that predicts the number of universal protein families given the observed data*
- 2. Focus on neighboring genes to the sulfur genes that were used to identify the vMAGs (per-gene analysis). These sets of genes are probably more complete. Therefore high diversity in these groups can be claimed to demonstrate high genomic variation.*
- 3. Provide some other analysis that will show that the observed diversity is due to high genomic diversity.*

Authors' Reply:

To address this comment we have performed additional analyses to respond to points 1 and 2 as suggested by the reviewer. Overall, our additional analyses back up our original claim that the mVCs, though only partial sequences, are diverse and have undergone evolution over time while retaining the AMGs.

For point 1, we have provided an additional analysis in order to validate the protein grouping as an accurate method to depict mVC diversity. In brief, we randomly selected 94 complete Caudovirales phages from NCBI RefSeq and compared the ratios and abundances of generated protein groups to those from the 94 mVC proteins grouped in the manuscript. We did this for 100 iterations. This analysis showed that our method to group proteins yielded comparable results to NCBI RefSeq phages, suggesting that we captured sufficient diversity. This analysis is provided in the Methods section (Methods lines 680-691, Results line 310).

For point 2 we performed the analysis suggested by the reviewer. From the 94 mVCs in the original protein grouping analysis, we extracted 5 proteins before and 5 proteins after the AMG. A total of 70 mVCs were used (some did not have 5 proteins before and after the AMG). The same protein grouping method for the whole mVCs was performed for these sets of 11 proteins. As suggested by the reviewer, high diversity in these neighboring proteins would indicate high diversity of the mVCs, regardless of only having partial sequences. Our results, provided as a heatmap supplementary figure, show that indeed the neighboring proteins are diverse (Methods line 673, Results lines 322-332).

REVIEWERS' COMMENTS

Reviewer #2 (Remarks to the Author):

The author responses and changes made in the manuscript satisfy my concerns. I have no further comments.